# Human DNA2 possesses a cryptic DNA unwinding activity that functionally integrates with BLM or WRN helicases

Cosimo Pinto[1], Kristina Kasaciunaite[2], Ralf Seidel[2], Petr Cejka[1]*

[1]Institute of Molecular Cancer Research, University of Zurich, Zurich, Switzerland;
[2]Institute of Experimental Physics I, University of Leipzig, Leipzig, Germany

**Abstract** Human DNA2 (hDNA2) contains both a helicase and a nuclease domain within the same polypeptide. The nuclease of hDNA2 is involved in a variety of DNA metabolic processes. Little is known about the role of the hDNA2 helicase. Using bulk and single-molecule approaches, we show that hDNA2 is a processive helicase capable of unwinding kilobases of dsDNA in length. The nuclease activity prevents the engagement of the helicase by competing for the same substrate, hence prominent DNA unwinding by hDNA2 alone can only be observed using the nuclease-deficient variant. We show that the helicase of hDNA2 functionally integrates with BLM or WRN helicases to promote dsDNA degradation by forming a heterodimeric molecular machine. This collectively suggests that the hDNA2 motor promotes the enzyme's capacity to degrade dsDNA in conjunction with BLM or WRN and thus promote the repair of broken DNA.

## Introduction

DNA replication, repair and recombination require the function of multiple DNA helicases and nucleases (*Tsutakawa et al., 2014*; *Wu and Hickson, 2006*). The DNA replication ATP-dependent helicase/nuclease 2 (DNA2) is an enzyme that contains both helicase and nuclease domains within the same polypeptide (*Bae et al., 1998*), and has important functions in a variety of DNA metabolic processes. Dna2 was first described in *Saccharomyces cerevisiae* where it is required for DNA replication under unperturbed conditions (*Budd and Campbell, 1995*; *Kuo et al., 1983*). Specifically, during Okazaki fragment processing, yeast Dna2 (yDna2) cleaves long 5'-flaps that are coated by the Replication Protein A (RPA) and are therefore refractory to cleavage by Rad27 (FEN1) (*Bae et al., 2001*; *Levikova and Cejka, 2015*). Moreover, yDna2 is one of the nucleases that resect 5'-terminated strands of DNA double-strand breaks (DSBs) (*Cejka et al., 2010*; *Niu et al., 2010*; *Zhu et al., 2008*). This process leads to the formation of 3'-tailed DNA, which becomes a substrate for the strand exchange protein Rad51 to initiate homology search and accurate DSB repair by the recombination machinery (*Cejka, 2015*; *Heyer et al., 2010*; *Symington, 2014*). Yeast Dna2 also functions upon replication stress to degrades structures such as reversed replication forks (*Hu et al., 2012*; *Thangavel et al., 2015*) and has a structural role in DNA damage signaling, where it is a component in one out of three signaling branches that activate the Mec1 kinase in response to ssDNA in S-phase (*Kumar and Burgers, 2013*). Additionally, yDna2 was described to be required for the proper function of telomeres (*Choe et al., 2002*). In contrast to Okazaki fragment processing and DNA end resection, the involvement of yDna2 in these latter DNA metabolic processes is poorly understood. The yeast Dna2 protein contains a large unstructured N-terminal domain, which mediates a physical interaction with yRPA (*Bae et al., 2003*), is required for Dna2's checkpoint function (*Kumar and Burgers, 2013*) and its capacity to melt secondary structures within 5' DNA flaps (*Lee et al., 2013*). The N-terminal domain is followed by a RecB-like nuclease domain

*For correspondence: cejka@
imcr.uzh.ch

Competing interests: The
authors declare that no
competing interests exist.

Reviewing editor: Antoine M
van Oijen, University of
Wollongong, Australia

(*Budd et al., 2000*) and a Superfamily I helicase domain in the C-terminal part of the polypeptide (*Budd and Campbell, 1995*). With the exception of checkpoint signaling, all Dna2 functions are exclusively dependent on its nuclease activity (*Sturzenegger et al., 2014*; *Thangavel et al., 2015*; *Wanrooij and Burgers, 2015*; *Zhu et al., 2008*). Dna2 homologs are present in all eukaryotic organisms including human cells (*Budd and Campbell, 1995*; *Eki et al., 1996*; *Gould et al., 1998*). Both helicase and nuclease domains are well conserved in evolution, but the unstructured N-terminal domain is only present in lower eukaryotes (*Bae et al., 1998*; *Kang et al., 2010*; *Wanrooij and Burgers, 2015*).

Human DNA2 (hDNA2) also functions in DNA end resection (*Gravel et al., 2008*; *Nimonkar et al., 2011*; *Sturzenegger et al., 2014*) and in the processing of non-canonical DNA replication structures, such as reversed replication forks upon replication stress (*Duxin et al., 2012*; *Thangavel et al., 2015*). In contrast to yeast, however, hDNA2 appears to be dispensable for the processing of most Okazaki fragments (*Duxin et al., 2012*). Specific inactivation of the nuclease, as well as the depletion or knockout of the protein/gene, result in lethal phenotypes in all organisms tested to date (*Budd et al., 2000*; *Duxin et al., 2012*; *Kang et al., 2000*; *Lin et al., 2013*). In yeast, this has been ascribed to yDna2's role in Okazaki fragment processing (*Kang et al., 2010*). Human DNA2-depleted cells arrest at late S/G2 phase of the cell cycle (*Duxin et al., 2012*). The nature of DNA intermediates that require the processing by hDNA2 is still rather elusive. It is conceivable that the lethality of hDNA2-depleted cells results from the failure to process reversed replication forks or other aberrant structures that arise during replication stress even in the absence of treatment with genotoxic drugs (*Duxin et al., 2012*; *Thangavel et al., 2015*). The role of hDNA2 in DSB end resection in contrast does not appear to be essential for viability as it functions redundantly with another nuclease, Exonuclease 1 (EXO1) (*Gravel et al., 2008*; *Nimonkar et al., 2011*; *Tomimatsu et al., 2012*). EXO1 is not involved in the processing of reversed replication forks, pointing towards an essential function of hDNA2 in the response to intermediates arising during DNA replication (*Thangavel et al., 2015*).

The nuclease of hDNA2 is specific for ssDNA (*Kim et al., 2006*; *Masuda-Sasa et al., 2006*) and therefore requires an associated helicase activity to resect/degrade dsDNA. This was shown to be either BLM or WRN during DSB end resection (*Gravel et al., 2008*; *Nimonkar et al., 2011*; *Sturzenegger et al., 2014*), or primarily WRN to degrade non-canonical DNA structures arising during DNA replication (*Thangavel et al., 2015*). Interestingly, the inherent helicase of hDNA2 was not required for these processes (*Sturzenegger et al., 2014*; *Thangavel et al., 2015*), and the function of the hDNA2 motor activity remains unclear. The helicase function is not essential for viability in yeast (*Bae et al., 2002*), where it was proposed to unwind secondary structures forming on long flaps at the 5' ends of Okazaki fragments (*Lee et al., 2013*). Yeast *dna2* cells lacking the helicase activity are dramatically sensitive to alkylating agents such as methyl methanesulfonate (MMS) (*Budd and Campbell, 1995*), suggesting that the yDna2 helicase might also play a role in the response to replication stress. In contrast to yeast, both helicase and nuclease functions are essential for viability in human cells (*Duxin et al., 2012*). Similarly to hDNA2 nuclease-deficient cells, hDNA2 helicase-deficient cells also exhibit a terminal S/G2 cell cycle arrest, most likely due to the inability to resolve structures arising in S-phase (*Duxin et al., 2012*). Furthermore, hDNA2 nuclease-deficient cells displayed cell cycle defects that were even more severe than upon depletion of hDNA2; interestingly, this phenotype was dependent on the integrity of the Walker A motif within the helicase domain (*Duxin et al., 2012*). This suggested that the hDNA2 helicase performs essential functions during DNA replication, yet it becomes toxic in the absence of the nuclease (*Duxin et al., 2012*), although mechanistic insights into the interplay between both activities have been lacking. Therefore, it remains to be determined how the hDNA2 helicase contributes to the overall function of the polypeptide.

The clear requirement for the helicase of hDNA2 for the viability of human cells (*Duxin et al., 2012*) stands in contrast to the inconclusive reports regarding the capacity of the human recombinant hDNA2 polypeptide to unwind dsDNA. One work concluded that hDNA2 lacks a helicase activity (*Kim et al., 2006*), whereas another study could detect DNA unwinding, albeit very weak and distributive (*Masuda-Sasa et al., 2006*). It has been also proposed that the helicase domain may be more responsible for DNA binding rather than as a motor activity per se (*Zhou et al., 2015*). Here we present that hDNA2 possesses a processive helicase activity capable of unwinding dsDNA of several kilobases in length. Paradoxically, the helicase is cryptic and becomes detectable only upon

inactivation of the nuclease. This explains the more pronounced phenotypes of the hDNA2 nuclease-deficient cells as opposed to double nuclease- and helicase-deficient cells or depletions of the polypeptide (*Duxin et al., 2012*). Finally, we show that the helicase of hDNA2 contributes to dsDNA degradation in complex with Bloom syndrome protein (BLM) or Werner syndrome protein (WRN) helicases, and may play a supporting role in the resection of DSBs or other aberrant structures arising during DNA replication. The motor activities within hDNA2 and BLM/WRN function in a synergistic manner, and the stimulatory effect observed with the hDNA2-WRN and hDNA2-BLM pairs is highly specific. This shows that the hDNA2-BLM and hDNA2-WRN complexes are functionally more integrated molecular machines than previously thought.

## Results

### Expression and purification of human DNA2

Human DNA2 was prepared using a construct, which contained an N-terminal 6x-histidine and a C-terminal FLAG affinity tags (*Figure 1A*). The sequence of hDNA2 was codon-optimized (*Supplementary file 1A*) for the expression in *Spodoptera frugiperda 9 (Sf9)* cells, which improved the yield ~2–3 fold (data not shown). Considering that hDNA2 contains an iron-sulfur cluster (*Pokharel and Campbell, 2012*; *Yeeles et al., 2009*), all buffers were degassed and contained reducing agents throughout the preparation procedure to prevent oxidation of the cluster, as described previously for *S. cerevisiae* Dna2 (*Levikova et al., 2013*). Wild type hDNA2, nuclease-deficient D277A, helicase-deficient K654R as well as nuclease- and helicase-deficient D277A K654R variants were purified in the same manner to near homogeneity (*Figure 1B* and *Figure 1—figure supplement 1A–C*). The yield of the recombinant proteins was ~330–390 µg from 3 liters of *Sf9* cell culture except for the variant containing the K654R mutation, which yielded only ~27 µg.

### hDNA2 preferentially degrades 5'-tailed DNA in the presence of RPA

Human DNA2 is known to possess ssDNA-specific nuclease activity (*Kim et al., 2006*; *Masuda-Sasa et al., 2006*). Considering that hDNA2 performs multiple functions during DNA metabolism, we set out to analyze the preference of its nuclease activity using various oligonucleotide-based DNA structures. Without the human Replication Protein A (hRPA), hDNA2 most efficiently degraded ssDNA, while 5'-overhanged, 3'-overhanged and Y-structured DNA were degraded ~7–20-fold less efficiently, based on the hDNA2 concentration required for the degradation of 50% DNA substrate (*Figure 1C* and *Figure 1—figure supplement 1D*). In contrast, dsDNA was largely refractory to cleavage (*Figure 1C* and *Figure 1—figure supplement 1D*), in agreement with the observations that hDNA2 needs a helicase partner in DNA end resection to initiate homologous recombination (*Cejka et al., 2010*; *Gravel et al., 2008*; *Nimonkar et al., 2011*; *Sturzenegger et al., 2014*; *Zhu et al., 2008*). As reported previously (*Masuda-Sasa et al., 2006*), the helicase-deficient hDNA2 (K654R) variant displayed a nuclease activity indistinguishable from that of the wild type enzyme on a 5'-tailed DNA substrate (*Figure 1—figure supplement 1D–F*). Using a 3'-end labeled ssDNA, we observed that hRPA directs the nuclease of hDNA2 towards the 5' terminus; while at the same time inhibits the 3'-5' nuclease activity (*Figure 1D*). This is in agreement with previous observations in various organisms (*Cejka et al., 2010*; *Nimonkar et al., 2011*; *Zhou et al., 2015*) and explains how hRPA enforces the correct polarity of DNA degradation during DNA end resection. Interestingly, in the presence of hRPA, hDNA2 most efficiently cleaved Y-structured and 5'-tailed DNA substrates, which were degraded ~5–10-fold more efficiently than ssDNA (*Figure 1E* and *Figure 1—figure supplement 1G*). In summary, the nuclease activities of yeast Dna2 and human DNA2 are very similar qualitatively, but human DNA2 appears somewhat less active (~2-fold in degradation of 5'-tailed DNA) than its yeast homologue (*Levikova et al., 2013*).

The nuclease-deficient hDNA2 D277A was subsequently used to determine DNA binding preference. hDNA2 D277A strongly bound ssDNA, with $K_D$ ~2 nM for ssDNA of 50 nucleotides in length (*Figure 1F,G*). Similar binding affinity was observed for Y-structured DNA, while the apparent DNA binding to 5' and 3'-tailed structures was reduced ~8–12-fold, respectively, compared to ssDNA. In contrast, dsDNA was bound very poorly (*Figure 1G* and *Figure 1—figure supplement 2A–D*). Further experiments revealed that the DNA binding affinity was determined by the length of ssDNA rather than the specific structure (*Figure 1H,I* and *Figure 1—figure supplement 2A–F*).

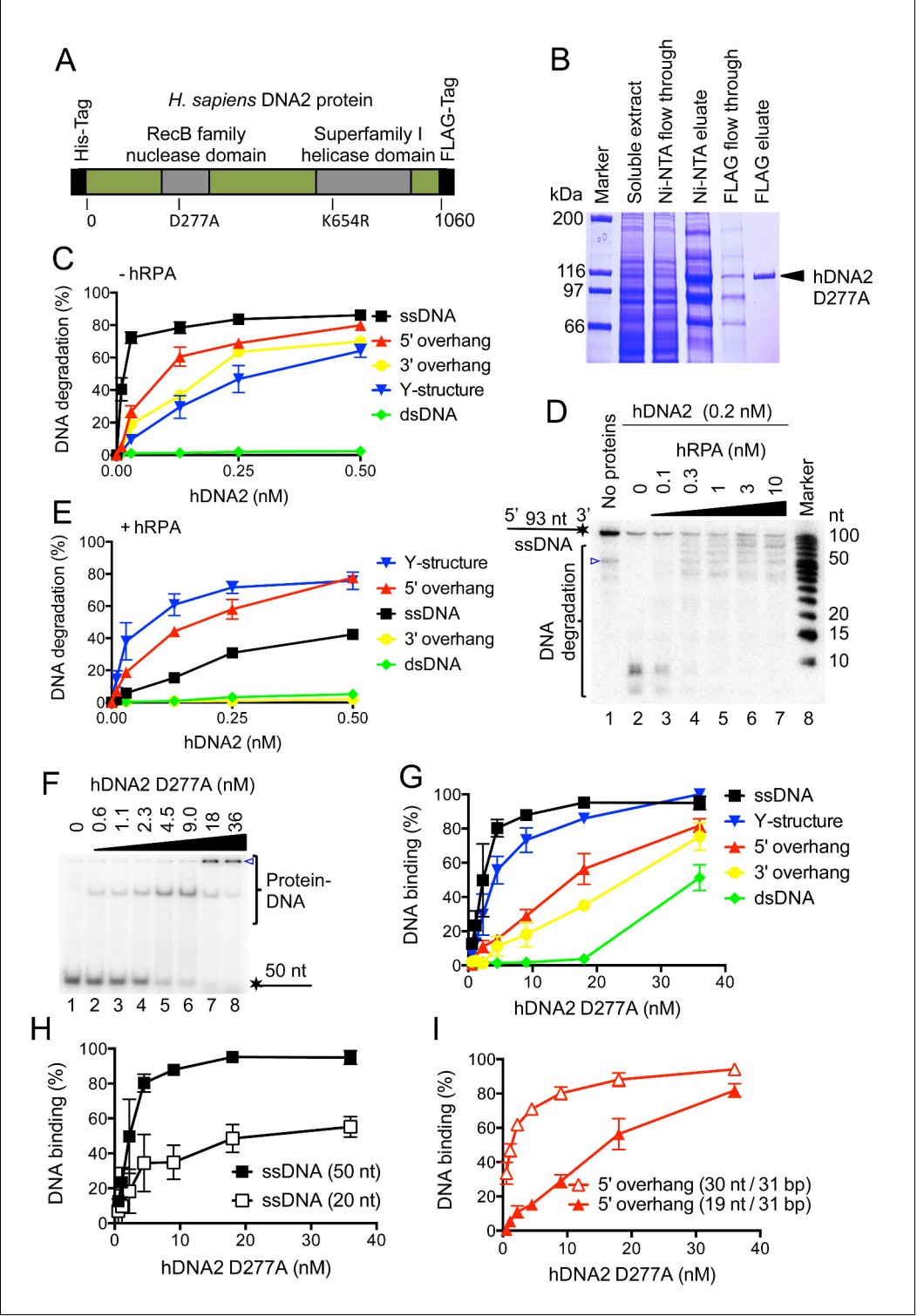

**Figure 1.** Human DNA2 preferentially binds and degrades 5′ terminated ssDNA. (**A**) A schematic representation of the recombinant hDNA2 protein used in this study. The polypeptide contains an N-terminal 6xHis- and a C-terminal FLAG affinity tag. The positions of the mutations inactivating the nuclease (D277A) activity or the helicase (K654R) activity are indicated. (**B**) A 10% polyacrylamide gel stained with Coomassie blue showing fractions from a representative purification of hDNA2 D277A. (**C**) Quantitation of hDNA2 nuclease activity on various DNA substrates in the absence of hRPA from experiments such as shown in *Figure 1—figure supplement 1D*. Averages shown, n = 2; error bars, SEM. (**D**) Human DNA2 (0.2 nM) was incubated with ssDNA [32]P-labeled at

*Figure 1 continued on next page*

*Figure 1 continued*

its 3' end and various concentrations of hRPA. The panel shows a representative denaturing 20% polyacrylamide gel. The blue triangle indicates a truncation of the substrate. (E) Quantitation of hDNA2 nuclease activity on various DNA substrates in the presence of hRPA (15 nM) from experiments such as shown in *Figure 1—figure supplement 1G*. Averages shown, n = 2; error bars, SEM. (F) A representative 6% polyacrylamide gel showing the binding of hDNA2 D277A to ssDNA of 50 nt in length. The blue triangle indicates the position of the wells. (G) Quantitation of DNA binding from experiments such as shown in *Figure 1F* and *Figure 1—figure supplement 2A–D*. Averages shown, n = 2–3; error bars, SEM. (H) DNA binding and its dependence on the length of ssDNA. Quantitation is based on experiments such as shown in *Figure 1F* and *Figure 1—figure supplement 2E*. Long ssDNA was more efficiently bound by hDNA2. Averages shown, n = 2–3, error bars, SEM. (I) DNA binding and its dependence on the length of 5' single-stranded DNA overhang. Quantitation is based on experiments such as shown in *Figure 1—figure supplement 2B,F*. Averages shown, n = 3; error bars, SEM.

The following figure supplements are available for figure 1:

**Figure supplement 1.** Human RPA guides the hDNA2 nuclease to 5' terminated ssDNA.

**Figure supplement 2.** hDNA2 binds ssDNA.

---

Interestingly, the hDNA2-bound DNA species either entered the polyacrylamide gels during electrophoresis, or remained stuck in the wells, indicative of a multiprotein-DNA complex and most likely a non-specific aggregate. Remarkably, the distinct DNA-protein species that entered the polyacrylamide gel were only observed with substrates containing a free 5' end such as Y-structured, 5' tailed or ssDNA substrates (*Figure 1F* and *Figure 1—figure supplement 2A,B,F*), suggesting that hDNA2 exhibits a preference for this structure even in the absence of hRPA. This likely reflects its role in 5' DNA end degradation in various metabolic processes (*Kang et al., 2010*; *Nimonkar et al., 2011*; *Sturzenegger et al., 2014*; *Zheng et al., 2008*).

## hDNA2 shows DNA structure-dependent ATPase activity

Previous reports concluded that hDNA2 hydrolyses ATP, as expected from a protein containing an SFI helicase domain (*Budd and Campbell, 1995*). We next determined the ATP hydrolysis rate of nuclease-deficient hDNA2 D277A in the presence of various DNA structures. The ATPase activity was strongly enhanced in the presence the DNA cofactors. The greatest stimulation, ~13-fold, was observed with 5'-tailed DNA (*Figure 2A*), in agreement with the 5'-3' polarity of the hDNA2 helicase (*Balakrishnan et al., 2010a*; *Balakrishnan et al., 2010b*; *Masuda-Sasa et al., 2006*). The apparent turnover rate ($k_{cat}$) of the ATP hydrolysis in the presence of 5'-tailed substrates of different lengths was $6.9 \pm 1.1$ s$^{-1}$ and $6.2 \pm 0.9$ s$^{-1}$. In contrast, dsDNA stimulated the hDNA2 ATPase to the lowest extent, ~four-fold, compared to reactions without DNA. Next we performed the ATPase assays in the presence of various amounts of poly(dT) DNA, which is a ssDNA devoid of any secondary structure. As expected, the ATP hydrolysis rate increased with poly(dT) concentration. The measured reaction rate values were fitted into a Michaelis-Menten curve with $V_{max} = 3.1 \pm 0.3$ µM·min$^{-1}$ and $K_M = 115 \pm 45$ nM (in nucleotides, *Figure 2B*), which corresponds to $k_{cat} = 4.3 \pm 0.4$ s$^{-1}$. The nuclease-deficient DNA2 D277A variant was used for the above assays, as the nuclease of wild type DNA2 interferes with its capacity to hydrolyze ATP by degrading DNA that serves as a co-factor of the ATPase activity. As demonstrated in *Figure 2C*, the rate of ATP hydrolysis by the nuclease-deficient D277A variant incubated with 5'-tailed substrate was constant over time. In contrast the rate of ATP consumption decreased quickly in case of wild type DNA2 (*Figure 2C*). We believe that the nuclease activity of hDNA2 rapidly degrades the 5' ssDNA overhang, producing a substrate that is less efficient as a cofactor for the ATPase activity. Very similar behavior was previously observed with yeast Dna2 (*Levikova et al., 2013*). Collectively, these experiments establish that the ATPase activity of hDNA2 qualitatively resembles that of the yeast Dna2 homologue in terms of DNA substrate preference and interplay with the nuclease activity, but it is ~10-fold less active in quantitative terms (*Levikova et al., 2013*).

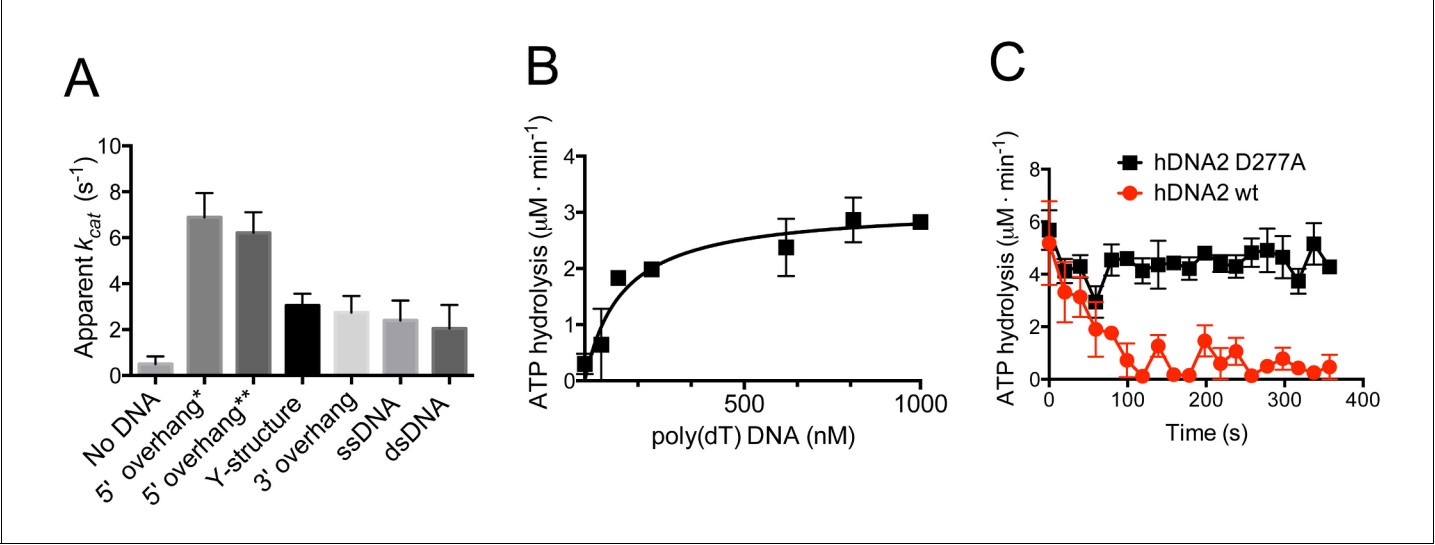

**Figure 2.** hDNA2 D277A shows DNA structure-dependent ATPase activity. (**A**) Apparent ATP turnover number $k_{cat}$ with various DNA cofactors, including short 5' overhang* (19 nt/31 bp), long 5' overhang** (30 nt/31 bp), Y-structure (19 nt/ 31 bp), 3' overhang (19 nt/ 31 bp), ssDNA (50 nt), dsDNA (31 bp). The reactions contained 12 nM hDNA2 D277A. Averages shown, n = 2–8; error bars, SEM. (**B**) Rate of ATP hydrolysis and its dependence on the DNA substrate concentration. The reactions contained 12 nM hDNA2 D277A and the indicated concentrations of poly(dT) DNA. Averages shown, n = 2; error bars, SEM. (**C**) Wild type hDNA2 or the D277A variant (both 12 nM) was incubated with 5' overhang DNA substrate and the rate of ATP hydrolysis was determined over time. The ATP hydrolysis rate was constant at ~4–5 $\mu M \cdot min^{-1}$ for hDNA2 D277A and decreased over time for wild type hDNA2. Averages shown, n = 3; error bars, SEM.

## The helicase of hDNA2 is capable to unwind plasmid-length dsDNA substrates

The capacity to unwind DNA by human DNA2 has been controversial. Previously, hDNA2's helicase activity was either described as undetectable (*Kim et al., 2006*) or very weak (*Balakrishnan et al., 2010a*; *Masuda-Sasa et al., 2006*), capable to unwind only short duplexes. In our earlier studies, we could show that *S. cerevisiae* Dna2 possesses a vigorous and processive helicase activity that is masked by its nuclease activity (*Levikova et al., 2013*). To this point, we set out to test whether our preparation of hDNA2 is capable to unwind dsDNA. The nuclease-deficient hDNA2 D277A variant was used in these experiments, as the nuclease activity of wild type hDNA2 may mask its helicase activity similarly as for the yeast enzyme (*Levikova et al., 2013*). We incubated various concentrations of the hDNA2 D277A variant with bacteriophage λDNA that had been digested with HindIII, resulting in dsDNA fragments of various lengths. *Figure 3A and B* demonstrate that hDNA2 D277A efficiently unwound dsDNA fragments of up to 2.3 kbp in length, whereas unwinding of ≥9.4 kbp-long fragments was barely detectable within the 30 min incubation time. Unwinding of these long DNA molecules was however evident in kinetic experiments upon longer incubation times (*Figure 3C* and *Figure 3—figure supplement 1A*). This unexpected dsDNA unwinding capacity of hDNA2 D277A was fully dependent on hRPA and ATP (*Figure 3A*). The nuclease-deficient hDNA2 D277A variant also similarly unwound a 2.7 kbp-long plasmid-based dsDNA substrate in a concentration-dependent manner (*Figure 3D,E*). In contrast, the nuclease- and helicase-deficient DNA2 D277A K654R mutant did not unwind DNA, as expected, showing that the unwinding capacity is inherent to the helicase activity of hDNA2 (*Figure 3—figure supplement 1B*). The above assays require the use of a high hRPA concentration to fully saturate DNA (~200–350 nM range), which can lead to dsDNA melting in the vicinity of the ends (*Georgaki and Hübscher, 1993*; *Kemmerich et al., 2016*). This might provide hDNA2 with ssDNA overhangs that are required for the unwinding activity. To define substrate preference for the hDNA2 D277A helicase, we next used a variety of oligonucleotide-based DNA structures. The hRPA concentration (7.5 nM) used in these assays did not result in a significant dsDNA melting. We found that the Y-structure was unwound most efficiently (*Figure 3F,G*), followed by the 5' overhang (*Figure 3G* and *Figure 3—figure*

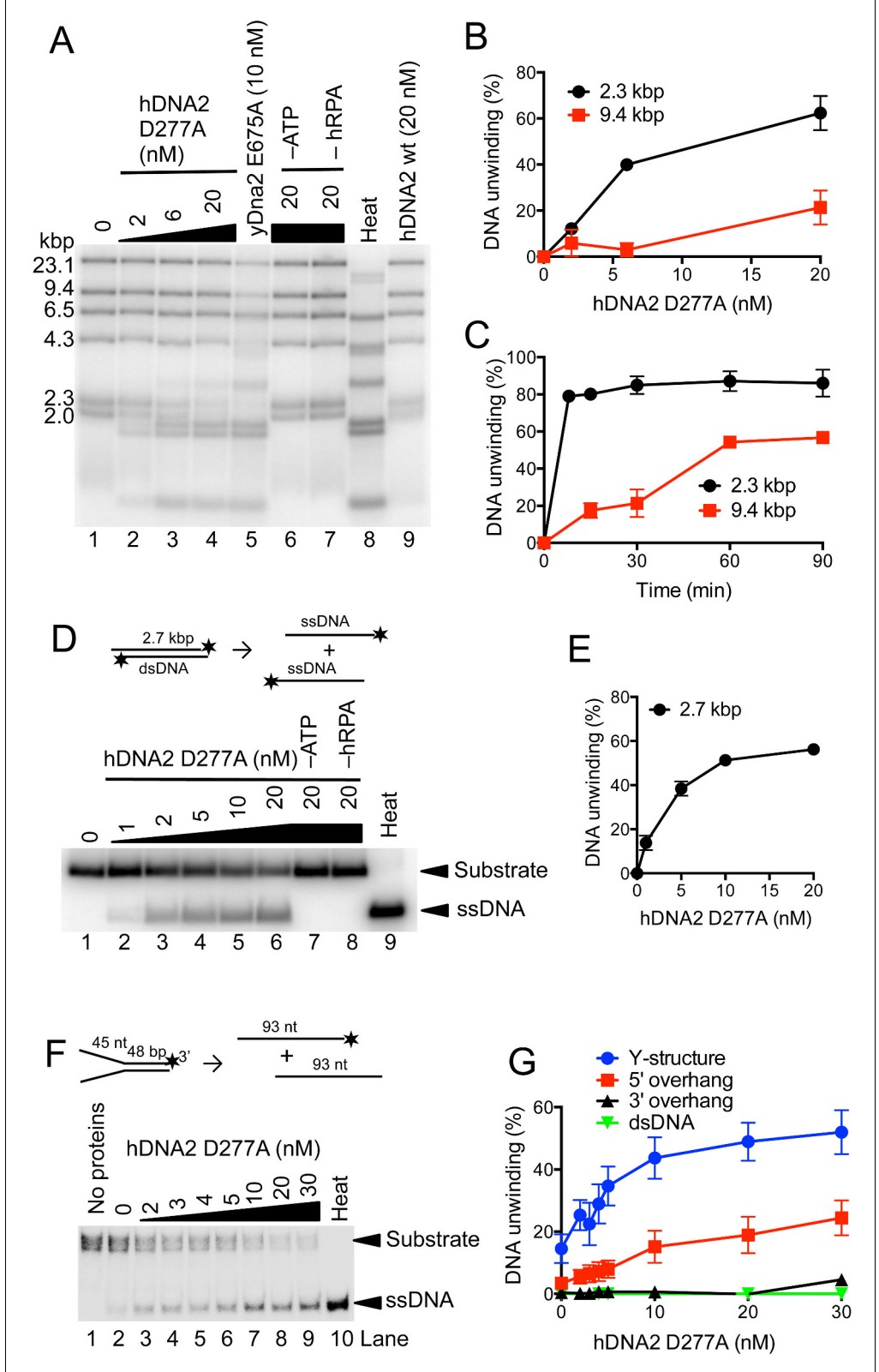

**Figure 3.** hDNA2 D277A unwinds kilobase-lengths of dsDNA. (**A**) Representative 1% agarose gel showing the helicase activity of hDNA2 D277A on λDNA/HindIII substrate with 346 nM hRPA. DNA unwinding leads to products that co-migrate with heat-denatured substrate (Lane 8). Lane 5, helicase activity of nuclease-deficient yeast Dna2 E675A at 30°C; Lane 6, no ATP; Lane 7, no RPA; Lane 8, heat-denatured DNA substrate; Lane 9, wild

*Figure 3 continued on next page*

*Figure 3 continued*

type hDNA2. (**B**) Unwinding of selected λDNA/HindIII fragments by various concentrations of hDNA2 D277A upon 30 min reaction time. Quantitation of experiments such as shown in *Figure 3A*. Averages shown, n = 2–4; error bars, SEM. (**C**) Unwinding of selected λDNA/HindIII fragments by hDNA2 D277A (20 nM) and its dependence on reaction time. Quantitation of experiments such as shown in *Figure 3—figure supplement 1A*. Averages shown, n = 2–4; error bars, SEM. (**D**) Representative 1% agarose gel showing the helicase activity of hDNA2 D277A on a 2.7 kbp-long substrate. Reactions contained 215 nM RPA. Heat, heat-denatured DNA substrate. (**E**) Quantitation of experiments such as shown in *Figure 3D*. Averages shown, n = 4–9; error bars, SEM. (**F**) Representative 10% polyacrylamide gel showing the helicase activity of hDNA2 D277A on an oligonucleotide-based Y-structure (45 nt/ 48 bp). Reactions contained 7.5 nM RPA. Heat, heat-denatured DNA substrate. (**G**) Quantitation of experiments such as shown in *Figure 3F* and *Figure 3—figure supplement 1C–E*. Beside Y-structure (45 nt/48 bp), DNA substrates with 5' or 3' overhangs (both 45 nt/ 48 bp) and blunt-ended dsDNA (50 bp) were tested. Reactions contained 7.5 nM RPA. Averages shown, n = 2–4; error bars, SEM. Heat, heat-denatured DNA substrate.

The following figure supplement is available for figure 3:

**Figure supplement 1.** hDNA2 D277A unwinds plasmid- and oligonucleotide-based DNA substrates.

---

*supplement 1C*). In contrast, no DNA unwinding was observed with 3' overhang and dsDNA (*Figure 3G* and *Figure 3—figure supplement 1D,E*), in agreement with the 5'-3' polarity of DNA unwinding by hDNA2 and its homologues (*Bae et al., 1998*).

Importantly, no dsDNA unwinding was observed with wild type hDNA2 (*Figure 3A*, lane 9). The hDNA2 nuclease is efficient in ssDNA degradation at sub-nanomolar concentrations (*Figure 1E*), and thus likely degrades the 5' ssDNA overhangs that are required for the initiation of DNA unwinding. By degrading 5'-tailed DNA the nuclease of hDNA2 masks the helicase capacity of the wild type polypeptide, similarly as in yeast (*Levikova et al., 2013*). This is in agreement with a recent structural study, which determined that the nuclease of DNA2 is situated ahead of the helicase domain, and therefore has the capacity to degrade 5'-terminated ssDNA tails to prevent DNA loading of the helicase domain (*Zhou et al., 2015*). Our observation that the inactivation of the hDNA2 nuclease unleashes the hDNA2 helicase likely provides explanation for the pronounced toxicity of a nuclease-deficient hDNA2 construct *in vivo*, which was dependent on the integrity of the Walker A motif within hDNA2 (*Duxin et al., 2012*). In summary, we show that nuclease-deficient hDNA2 possesses the capacity to unwind dsDNA of kilobases in length in a reaction dependent on hRPA and ATP. Wild type hDNA2 is devoid of any apparent dsDNA unwinding activity, which likely infers the existence of mechanisms that allow the manifestation of the motor activity in the context of the wild type polypeptide under specific conditions.

## DNA unwinding by hDNA2 D277A is slow but highly processive

DNA unwinding experiment with λDNA (*Figure 3A–C* and *Figure 3—figure supplement 1A*) showed that hDNA2 D277A can unwind long stretches of dsDNA. We next performed a kinetic experiment with a 2.7 kbp-long DNA substrate, and compared the DNA unwinding of the nuclease-deficient human DNA2 D277A and the yeast Dna2 E675A. As DNA2 cannot unwind DNA from internal sites and can only initiate from a free DNA end (*Balakrishnan et al., 2010a*; *Zhou et al., 2015*), we used a 20-fold excess of each helicase over the substrate to saturate the DNA ends. We detected ssDNA after only one minute of the reaction with the yeast enzyme, whereas it took eight minutes to detect a similar amount of unwound DNA for the human DNA2 variant (*Figure 3—figure supplement 1F,G*). To better define the unwinding rate and processivity of the hDNA2 D277A helicase, we applied a single molecule unwinding assay based on magnetic tweezers. We used a 5' tailed dsDNA substrate of 6.1 kbp in length, which was tethered at one end to the surface of a fluidic cell and at the other end to a magnetic bead. An externally applied magnetic field gradient allowed thus to hold the DNA in a stretched configuration. DNA unwinding (*Levikova et al., 2013*) was monitored by a change of the position of the magnetic bead as a result of different lengths of double- and single-stranded DNA (*Figure 4A*). We observed that hDNA2 unwound the dsDNA substrate slowly but consecutively over time (*Figure 4B*). Most of the DNA molecules underwent significant dsDNA unwinding of several kilobases in length, with some molecules showing unwinding of the full-length

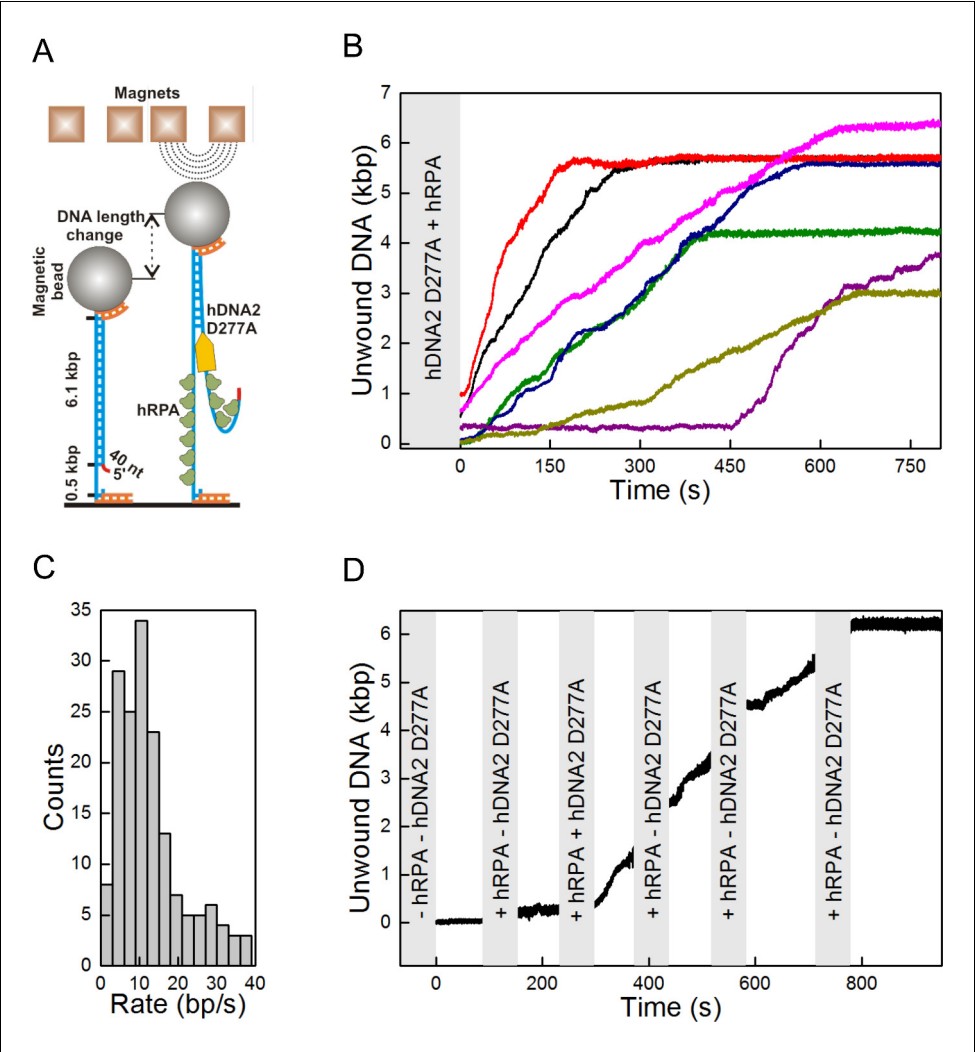

**Figure 4.** Single molecule experiments reveal highly processive DNA unwinding by hDNA2 D277A. (**A**) A sketch of the magnetic tweezers assay. (**B**) Representative DNA unwinding events (n = 7, colored) catalyzed by hDNA2 D277A at 22 ± 3 pN force. Experiments were conducted at 37°C in a reaction buffer supplemented with 25 nM hDNA2 D277A and 25 nM hRPA. DNA lengthening was observed only after the addition of hDNA2 D277A. (**C**) Histogram of the observed unwinding rates. Unwinding trajectories were divided into segments with approximately constant rate. The unwinding rates of the individual segments were determined from a linear fit of the data. (**D**) DNA unwinding experiment at 21 pN force, initiated by adding hRPA (25 nM) at 100 s and hDNA2 D277A (25 nM) at 220 s. The buffer containing hDNA2 D277A was washed away subsequently as indicated by the gray bars.

The following figure supplement is available for figure 4:

**Figure supplement 1.** Single molecule experiments reveal that DNA unwinding by hDNA2 D277A is dependent on ATP and hRPA.

6.1 kbp substrate. To quantify the unwinding rate, unwinding trajectories (*n* = 30) were split into successive segments that each had approximately a constant unwinding velocity. Unwinding rates were determined from the slope of a linear fit to each segment (*Figure 4C*). The unwinding rates were broadly distributed between 0 to 40 bp/s. Such a broad distribution is in agreement with measurements for yDna2 (*Levikova et al., 2013*). The distribution had a pronounced maximum at around 10 bp·s$^{-1}$ and the mean unwinding rate was 12.8 ± 0.8 bp·s$^{-1}$, which is ~3-fold slower than that of the yeast homologue (*Levikova et al., 2013*). Long-range processive dsDNA unwinding was dependent

on ATP and hRPA (*Figure 4—figure supplement 1A–D*), in agreement with the experiments shown above. To confirm that the observed dsDNA unwinding events resulted from the activity of a single hDNA2 D277A molecule, we performed an unwinding experiment similar that that in *Figure 4B*, but flushed in ATP and hRPA containing buffer at regular intervals (*Figure 4D*) such that any free hDNA2 D277A was removed. DNA unwinding continued despite successive washing steps, indicating that the originally acting unwinding complex remained bound and active during the course of the observation period. This demonstrated that the DNA2 D277A helicase is highly processive (*Figure 4D*). The recently-published structure of DNA2 shows that ssDNA must feed through a narrow tunnel to reach the helicase domain (*Zhou et al., 2015*), which likely prevents dissociation of DNA2 from its substrate and corroborates the high processivity observed in our assays. In agreement with this no direction reversals during unwinding (i.e. rezipping) that could originate from strand–switches were observed (*Dessinges et al., 2004*; *Klaue et al., 2013*).

## Regulation of hDNA2 nuclease and helicase activities by single-stranded DNA binding proteins

The hRPA protein is a critical cofactor of the hDNA2 nuclease (*Figure 1*) as well as the helicase activities (*Figure 3*) (*Nimonkar et al., 2011*; *Zhou et al., 2015*). We next set out to define the interplay of hDNA2 with other cognate ssDNA binding proteins. Initially, hDNA2 was described as a mitochondrial protein (*Zheng et al., 2008*) before its function in nuclear DNA metabolism was determined (*Duxin et al., 2009*; *Gravel et al., 2008*). Mitochondria are devoid of hRPA; instead, they contain the mitochondrial single-stranded DNA binding protein (mtSSB), a homotetramer similar to the SSB protein of *Escherichia coli* (*Curth et al., 1994*). Furthermore, the sensor of single-stranded DNA (SOSS) complex was described in the nucleus of human cells (*Huang et al., 2009*; *Li et al., 2009*). SOSS likely regulates various aspects of DNA metabolism including DNA recombination and DNA end resection by the MRE11-RAD50-NBS1 (MRN) complex and EXO1 (*Richard et al., 2011*; *Yang et al., 2013*). We purified both mtSSB and SOSS and ascertained that both complexes bind ssDNA with a high affinity (*Figure 5—figure supplement 1A–D*). Next we investigated whether dsDNA unwinding by hDNA2 D277A can be promoted by mtSSB or the SOSS complex similarly as by hRPA. Neither mtSSB (*Figure 5A–C* and *Figure 5—figure supplement 1E*) nor SOSS (*Figure 5C* and *Figure 5—figure supplement 1E*) were able to substitute hRPA to promote dsDNA unwinding. The hDNA2-hRPA functional interaction was largely species-specific, as yRPA from *S. cerevisiae* promoted DNA unwinding to a much lesser extent than the cognate hRPA (*Figure 5A–C* and *Figure 5—figure supplement 1E*). Therefore, hRPA is a unique and an essential co-factor of the hDNA2 helicase, which is required for the unwinding of all DNA duplex substrate lengths tested.

In contrast to dsDNA unwinding, hRPA could be replaced by yRPA in ssDNA degradation, which similarly directed the nuclease activity of hDNA2 towards the 5' end of ssDNA (*Figure 1D*, *Figure 5D* and *Figure 5—figure supplement 1F*). Similar, albeit much weaker effect was observed in the presence of the SOSS complex (*Figure 5—figure supplement 1F*, lane 28), possibly due to a lower affinity of SOSS towards ssDNA compared to the RPA proteins. Considering the postulated function of the hDNA2 nuclease in mitochondrial DNA metabolism, the mtSSB unexpectedly dramatically inhibited all nuclease activities of hDNA2 (*Figure 5D* and *Figure 5—figure supplement 1F*). Therefore it remains to be elucidated how the hDNA2 nuclease/helicase functions in mitochondria.

## The helicase of hDNA2 promotes DNA end resection in conjunction with WRN or BLM helicases

It has been established that hDNA2 functions in conjunction with a helicase partner in DNA end resection. Initially, it has been described that the cognate partner is BLM (*Gravel et al., 2008*; *Nimonkar et al., 2011*). Later, it was demonstrated that also WRN could function in a redundant manner instead of BLM, or even be the sole helicase partner of hDNA2 during resection of reversed replication forks (*Sturzenegger et al., 2014*; *Thangavel et al., 2015*). Having demonstrated that hDNA2 possesses a helicase activity, we wondered whether either BLM or WRN could stimulate hDNA2 or *vice versa*, i.e., whether the enzyme complex may form an integrated unit. To this point, we expressed wild type WRN and BLM helicases as well as their variants in *Sf9* insect cells and purified all polypeptides to near homogeneity (*Figure 6—figure supplement 1A–D*). We next monitored the resection of a 2.7 kbp-long dsDNA substrate. We selected wild type WRN and BLM

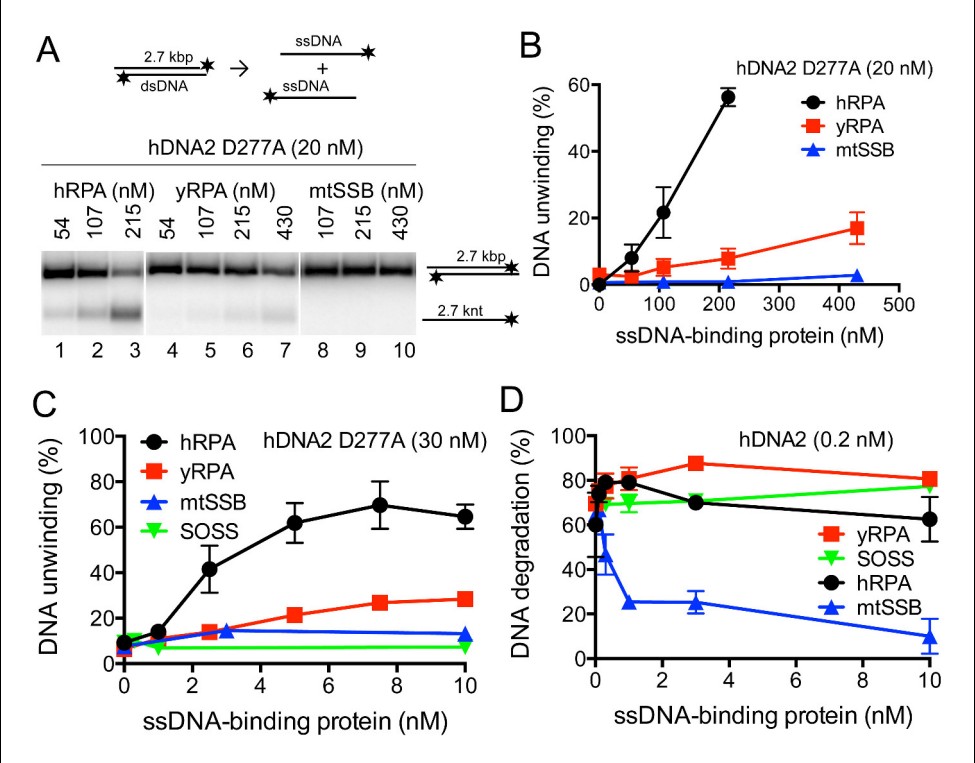

**Figure 5.** hDNA2 nuclease and helicase activities are regulated by ssDNA-binding proteins. (**A**) Representative 1% agarose gels showing the helicase activity of hDNA2 D277A supplemented with indicated ssDNA-binding proteins on a $^{32}$P-labeled 2.7 kbp-long dsDNA substrate. (**B**) Quantitation of experiments such as shown in *Figure 5A*. Averages shown, n = 3–9; error bars, SEM. (**C**) Quantitation of unwinding experiments with Y-structured oligonucleotide-based DNA such as shown in *Figure 5—figure supplement 1E*. Averages shown, n = 2; error bars, SEM. (**D**) Quantitation of ssDNA degradation from experiments such as shown in *Figure 1D* and *Figure 5—figure supplement 1F*. Averages shown, n = 2; error bars, SEM.

The following figure supplement is available for figure 5:

**Figure supplement 1.** hDNA2 nuclease and helicase activities are regulated by ssDNA-binding proteins.

concentrations that led to a partial unwinding of the substrate (*Figure 6A* lane 2, *Figure 6B*, lane 2). Titrating wild type or helicase-deficient (K654R) hDNA2 into these reactions led to the degradation of the unwound ssDNA, as expected. At the same time, the overall degradation of the dsDNA substrate increased as well (*Figure 6A–C*). Under the same conditions, hDNA2 did not degrade dsDNA without the helicase partner (*Figure 6A*, lane 11). This indicates a synergistic relationship between the two enzyme pairs, i.e. that dsDNA unwinding/degradation by the enzyme pair is up to six-fold higher than the sum of activities of the polypeptides acting individually (*Figure 6C*). Furthermore, wild type hDNA2 was more efficient in DNA degradation than its helicase-deficient variant, in particular together with the WRN helicase (*Figure 6A,C*). As shown above, both wild type and helicase-deficient hDNA2 variants had indistinguishable nuclease activity on 5'-labeled oligonucleotide based substrate (*Figure 1—figure supplement 1E,F*). We could observe a similar stimulatory effect when using a fixed concentration of hDNA2 and titrating either WRN or BLM helicases into the reactions (*Figure 6D,E,J*).

To this point, the experiments demonstrated that DNA degradation by hDNA2 was stimulated by a DNA helicase added in *trans*. To determine whether the stimulatory effect by WRN or BLM is specific for these two helicases, we tested if other human RecQ helicase family members could substitute WRN or BLM in the DNA end resection assays. We did not observe any enhancement of DNA degradation by hDNA2 upon adding either RecQ1 or RecQ5 helicases (*Figure 6—figure*

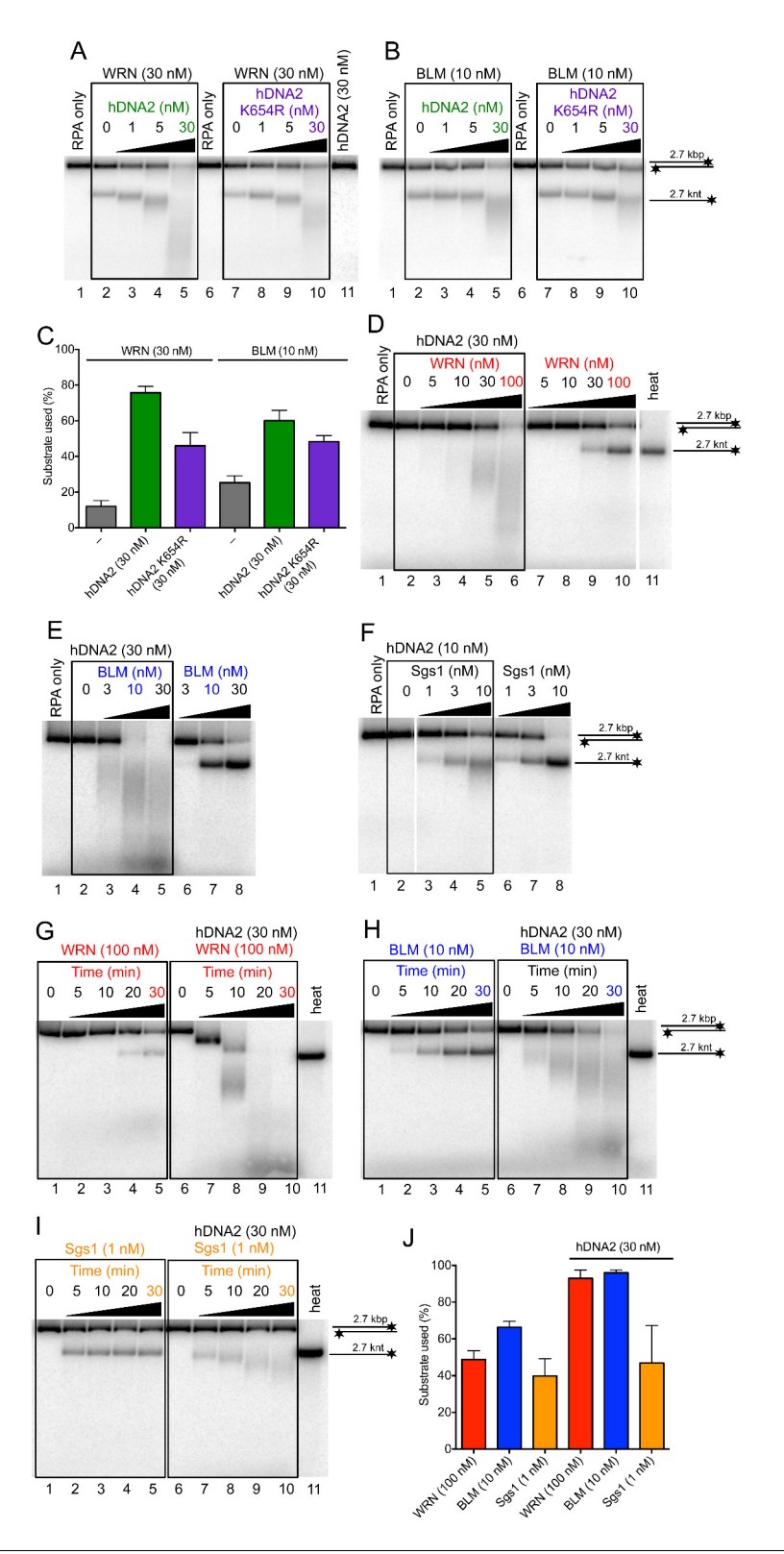

**Figure 6.** hDNA2 synergizes with WRN and BLM in the degradation of dsDNA. Representative 1% agarose gels showing dsDNA degradation by wild type or helicase-deficient hDNA2 K654R variant with (**A**) WRN or (**B**) BLM. The reactions were supplemented with 50 mM NaCl and 215 nM hRPA. (**C**) Quantitation of experiments such as shown in *Figure 6A,B*. Averages shown, n = 4–6; error bars, SEM. Representative 1% agarose gels showing dsDNA processing by hDNA2 and (**D**) WRN, (**E**) BLM or (**F**) yeast Sgs1. The reactions were supplemented with 50 mM NaCl and 215 nM hRPA. Representative

*Figure 6 continued on next page*

*Figure 6 continued*

1% agarose gels showing the kinetics of dsDNA processing by hDNA2 and (**G**) WRN (**H**) BLM and (**I**) yeast Sgs1. The reactions were supplemented with 50 mM NaCl and 215 nM hRPA. (**J**) Quantitation of experiments such as shown in *Figure 6D,E,G–I*. Averages shown, n = 3; error bars, SEM.

The following figure supplement is available for figure 6:

**Figure supplement 1.** Purification of WRN and BLM proteins.

*supplement 1E*). However, RecQ1 and RecQ5 did not show detectable unwinding of the 2.7 kbp-long dsDNA on their own. Therefore, we next used Sgs1, which shows a vigorous DNA helicase activity and functions in conjunction with yDna2 in resection (*Cejka et al., 2010*). Importantly, adding hDNA2 to reactions together with Sgs1 resulted in the degradation of unwound ssDNA, but in contrast to the reactions with BLM or WRN, no additional double-stranded substrate was degraded. Moreover, hDNA2 even appeared to inhibit dsDNA unwinding by Sgs1 (*Figure 6F*). The specific stimulatory effect was also observed in kinetic experiments with BLM and WRN, while no stimulation was observed with hDNA2 and Sgs1 (*Figure 6G–J*). The stimulation of DNA degradation by hDNA2 was particularly pronounced together with WRN, where a gradual degradation of the substrate was observed (*Figure 6G*). As our substrate is labeled on the 3' end, the observed degradation pattern is indicative of a 5'-3' polarity of DNA degradation by hDNA2, and appears unrelated to the 3'-5' exonuclease of WRN on recessed 3' ends (*Figure 6G*). Together, these experiments show that the hDNA2-BLM and hDNA2-WRN functionally integrate and that the helicase of hDNA2 may have a stimulatory role in DNA end resection.

## The WRN or BLM helicases functionally integrate with the helicase of hDNA2

Our previous work revealed that nuclease-deficient yDna2 E675A exhibits a very vigorous DNA helicase activity similar to the helicase of Sgs1, one of the most active helicases in eukaryotes (*Cejka and Kowalczykowski, 2010*; *Levikova et al., 2013*). Interestingly, we found here that under elevated ionic strength conditions, the yDna2 E675A variant was even more active than Sgs1 (*Figure 7A*). In contrast DNA unwinding by hDNA2 D277A was highly sensitive to NaCl, to a much greater extent than dsDNA unwinding by BLM or WRN (*Figure 7B*). Having established that hDNA2 with WRN or BLM synergize in DNA resection (*Figure 6*), we decided to determine the specific interplay of the helicases in more detail. At first, we tested whether helicase-deficient BLM K695A or helicase-deficient WRN K577M can stimulate DNA unwinding by nuclease-deficient hDNA2 D277A. At low salt concentrations (25 mM), the helicase activity of hDNA2 D277A is already strongly reduced (*Figure 7B*). Supplementing the reaction with WRN K577M or BLM K695A stimulated the hDNA2 D277A helicase activity ~7 or ~4-fold, respectively (*Figure 7C,D*). Therefore, both BLM and WRN have structural roles to promote dsDNA unwinding by the nuclease-deficient hDNA2 D277A. This stimulatory effect was specific for WRN and BLM helicases, as none of the other helicase-deficient enzymes tested stimulated DNA unwinding to a comparable extent (*Figure 7—figure supplement 1A–C*). At 50 mM NaCl, the unwinding by the nuclease-deficient hDNA2 D277A alone was completely inhibited (*Figure 7B,E–H*). Under the same conditions, we could observe that adding the hDNA2 D277A variant to dsDNA unwinding reactions containing wild type WRN resulted in an increase in DNA unwinding (~2.4-fold), which was more that upon the addition of the nuclease- and helicase-deficient hDNA2 D277A K654R variant (~1.5-fold stimulation, *Figure 7E,F*). Therefore, the hDNA2 helicase functionally integrates with WRN even under experimental conditions when no inherent helicase activity was detected. In contrast, we observed that hDNA2 has only a structural role to promote dsDNA unwinding by wild type BLM, as both hDNA2 D277A and DNA2 D277A K654R variants stimulated DNA unwinding by BLM to the same extent (*Figure 7G,H*), as noted by Sung and colleagues previously (*Daley et al., 2014*). Taken together, these data strongly suggest that the helicases of hDNA2 and WRN or BLM, respectively, function in an integrated manner, where one polypeptide stimulates the motor activity of its partner, in a mode reminiscent to that observed in prokaryotic resection machineries (*Dillingham and Kowalczykowski, 2008*). This is in agreement

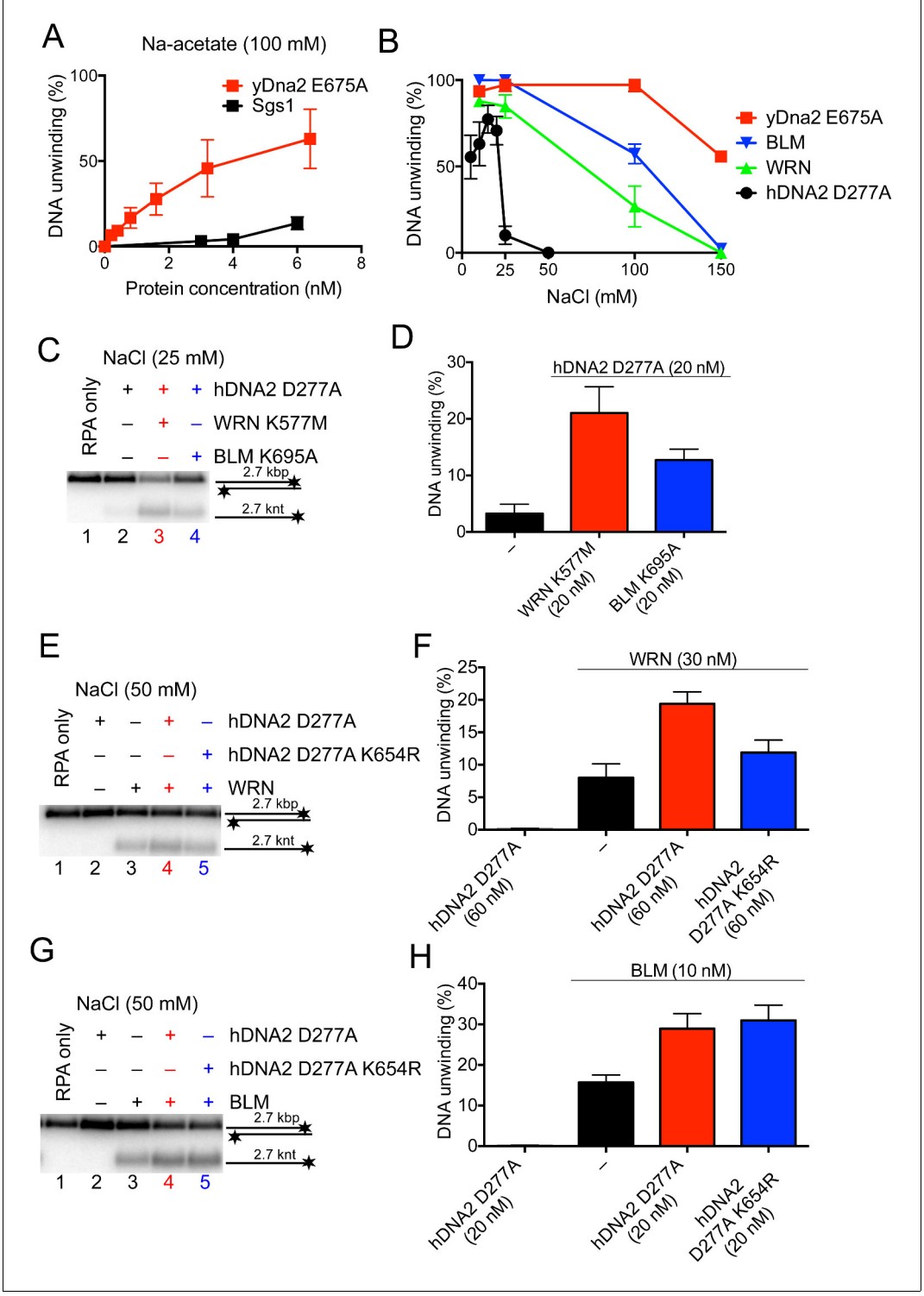

**Figure 7.** The helicase activity of hDNA2 functionally integrates with BLM or WRN helicases. (**A**) Quantitation of 2.7 kbp-long dsDNA unwinding by yDna2 E675A or Sgs1 with 400 nM of yeast RPA. Reactions were supplemented with 100 mM sodium acetate and 5 mM magnesium acetate and incubated at 30°C. Averages shown, n = 2–3; error bars, SEM (**B**) Quantitation of DNA unwinding by yDna2 E675A (1 nM), BLM (10 nM), WRN (30 nM), hDNA2 D277A (30 nM) and its dependence on NaCl concentration. Reactions were supplemented with indicated NaCl concentrations and 2 mM magnesium acetate and incubated at 37°C. Averages shown, n = 2–3; error bars, SEM. (**C**) Representative 1% agarose gel showing DNA unwinding by hDNA2 D277A (20 nM) and its stimulation by helicase-deficient WRN K577M (20 nM) and BLM K695A (20 nM) variants. The reactions were supplemented with

*Figure 7 continued on next page*

*Figure 7 continued*

25 mM NaCl and contained 215 nM hRPA. (**D**) Quantitation of experiments such as shown in *Figure 7C*. Averages shown, n = 5–7; error bars, SEM. (**E**) Representative 1% agarose gel showing the interplay of wild type WRN (30 nM) and nuclease-deficient hDNA2 D277A (60 nM) or nuclease- and helicase-deficient hDNA2 D277A K654R (60 nM) mutants. The reactions were supplemented with 50 mM NaCl and 215 nM hRPA. (**F**) Quantitation of experiments such as shown in *Figure 7E*. Averages shown, n = 3–4; error bars, SEM. (**G**) Representative 1% agarose gel showing the interplay of wild type BLM (10 nM) and nuclease-deficient hDNA2 D277A (20 nM) or nuclease- and helicase-deficient hDNA2 D277A K654R (20 nM) mutants. The reactions were supplemented with 50 mM NaCl and 215 nM hRPA. (**H**) Quantitation of experiments such as shown in *Figure 7G*. Averages shown, n = 2–4; error bars, SEM.

The following figure supplement is available for figure 7:

**Figure supplement 1.** The functional integration of the helicase activity of hDNA2 is specific for WRN and BLM.

with physical interactions between hDNA2 and BLM or WRN, respectively (*Nimonkar et al., 2011*; *Sturzenegger et al., 2014*).

## Discussion

The involvement of DNA2 in cellular metabolism is absolutely dependent on its nuclease activity in all organisms tested to date (*Kang et al., 2010*; *Wanrooij and Burgers, 2015*). The only exception appears to be *S. cerevisiae* Dna2's function in S-phase checkpoint signaling, where yeast Dna2 has a structural and not an enzymatic role (*Kumar and Burgers, 2013*). In all other cases, DNA2 mutants lacking the nuclease activity are as deficient as knockdowns and/or deletion mutants (*Budd et al., 2000*; *Duxin et al., 2012*; *Kang et al., 2000, 2010*; *Lin et al., 2013*; *Wanrooij and Burgers, 2015*). Indeed, DNA2 nuclease-deficiency brings about cellular lethality in yeast as well as human cells (*Budd et al., 2000*; *Duxin et al., 2012*; *Lee et al., 2000*). Much less is known about the function of the DNA2 helicase, despite both nuclease and helicase domains are equally conserved in evolution (*Bae et al., 1998*). Helicase-deficient yeast mutants are viable under most growth conditions (*Budd et al., 1995*; *Formosa and Nittis, 1999*), while in contrast the helicase appears to be essential in human cells (*Duxin et al., 2012*). Here we present that human DNA2 possesses a marked DNA unwinding activity (*Figure 3*), which can separate DNA duplexes of several kilobases in length in a reaction that is dependent on the presence of the single-strand DNA binding protein hRPA. Single molecule experiments using magnetic tweezers revealed that dsDNA unwinding by hDNA2 is highly processive (*Figure 4*). Together, this indicates that the motor activity of the polypeptide is conserved in evolution. Similarly to yeast (*Levikova et al., 2013*), the unwinding capacity of hDNA2 is cryptic as it is masked by the nuclease within the same polypeptide. Therefore, the dsDNA unwinding can only be observed with the nuclease-deficient hDNA2 D277A variant. We showed that the nuclease of hDNA2 degrades 5'-tailed DNA at subnanomolar concentrations. The hDNA2 D277A helicase requires 5'-terminated ssDNA strands for loading onto the DNA substrate; hence, we believe that the nuclease of wild type hDNA2 cleaves these 5' overhangs, which prevents loading of hDNA2 onto the substrate and the engagement of its motor activity. This observation is in agreement with the recently published structure of mouse DNA2, which demonstrated that the nuclease active site is located along the entrance of a narrow tunnel. In order for the 5' terminated DNA to reach the helicase domain, the DNA molecule must thread half way through the tunnel (*Zhou et al., 2015*). Thus, the position of the nuclease domain ahead of the helicase clearly explains the functional interplay we observed in our experiments. Interestingly, in knockdown-rescue experiments, the expression of the nuclease-deficient hDNA2 variant was much more detrimental than the expression of the double nuclease- and helicase-deficient polypeptide (*Duxin et al., 2012*). Duxin *et al.* proposed that the nuclease-deficient hDNA2 variant is likely toxic; our results that revealed that the inactivation of the nuclease unleashes the helicase of hDNA2, which, when uncontrolled, likely explains the cellular toxicity of the D277A mutant seen *in vivo* (*Duxin et al., 2012*).

The observation that the helicase of hDNA2 is only apparent upon inactivation of the nuclease raises the question about the physiological relevance of such motor activity, despite this interplay is

conserved from yeast to man (*Bae et al., 1998*; *Levikova et al., 2013*). Previously, we speculated that the interplay of the helicase and nuclease activities might be regulated by post-translational modifications, protein partners and/or specific DNA structures (*Levikova et al., 2013*). In such a scenario, e.g. a post-translational modification may selectively lower the nuclease activity, which could allow manifestation of the hDNA2 helicase. This may be also achieved by regulating the redox state of the hDNA2 iron-sulfur cluster, which is embedded in the nuclease domain, and the function of which remains unclear (*Pokharel and Campbell, 2012*; *Wu and Brosh, 2012*). Likewise, it is possible that protein partners such as BLM or WRN may functionally integrate with the enzymatic activities of hDNA2, and modulate the interplay of the helicase-nuclease within the polypeptide. The helicase of hDNA2 might therefore only engage when hDNA2 functions in complex with the BLM or WRN factors. Interestingly, the quantitative comparison of yDna2 and hDNA2 unwinding capacities revealed that the human homologue is approximately 3–10-fold slower than its yeast counterpart. Moreover, also human BLM or WRN appear to be about an order of magnitude less active than yeast Sgs1, the partner of yDna2 and the most active RecQ helicase identified to date (*Cejka and Kowalczykowski, 2010*; *Gray et al., 1997*; *Janscak et al., 2003*; *Karow et al., 1997*). These results may suggest that the two motor activities of yDna2/hDNA2 and Sgs1/BLM/WRN might have co-evolved to match the speed of each other's partner, and that Sgs1-yDna2 and BLM-hDNA2 or WRN-hDNA2 pairs might operate as functional units. Indeed, we observed here that adding hDNA2 to reactions containing BLM or WRN, hDNA2 not only degraded the unwound ssDNA, but the concerted activity of the enzyme pair resulted in an overall stimulation of the dsDNA degradation activity. This effect was specific for the hDNA2-WRN and hDNA2-BLM enzyme pairs, as no such stimulation was observed together with Sgs1. These hDNA2-WRN and hDNA2-BLM resection complexes might be the functional analogs of the DNA-end processing machineries in prokaryotes. In most gram-negative bacteria such as *E. coli*, the RecBCD complex consists of subunits that function autonomously but integrate into a molecular machine that has helicase-nuclease activities exceeding the sum of its parts. The RecB subunit contains a 3'-5' helicase (therefore, opposite to DNA2) followed by a dual polarity nuclease, which integrates with the 5'-3' motor of the RecD subunit (*Dillingham et al., 2003*; *Spies et al., 2007*; *Taylor and Smith, 2003*). In gram-positive bacteria such as *Bacillus subtilis*, the AddAB complex has two nucleases but only one helicase (*Rocha et al., 2005*; *Yeeles and Dillingham, 2007*). Similarly to DNA2, the AddB subunit contains an iron-sulfur cluster (*Yeeles et al., 2009*), and the structure of the nuclease domain shows a high level of similarity with mouse DNA2 (*Krajewski et al., 2014*; *Zhou et al., 2015*). Furthermore, most bacteria also contain the RecQ-RecJ complex (*Morimatsu and Kowalczykowski, 2014*; *Persky and Lovett, 2008*), which likewise provides complementary activities that integrate within a complex capable to resect DNA for the RecF recombination pathway. Having uncovered the cryptic helicase capacity within hDNA2, we sought to determine whether it might function synergistically together with the BLM and/or the WRN helicase. The results presented here clearly show that although both wild type and helicase-deficient hDNA2 variants have the same nuclease activity on 5'-tailed DNA, the wild type enzyme is clearly more proficient in DNA end resection under limiting enzyme concentrations (*Figure 6*). This indicated that the motor activity within nuclease-proficient hDNA2 contributes to DNA degradation in reactions containing WRN or BLM helicases. We believe that BLM and WRN helicases provide the lead motor activity, while the hDNA2 motor has an accessory function, possibly to enhance the processivity of the complex, to help traverse strand discontinuities or to degrade unwound ssDNA. Neither WRN nor BLM could inhibit ssDNA degradation by hDNA2 (data not shown), so we do not believe that WRN/BLM's function to facilitate the engagement of hDNA2 motor activity results from an inhibition of its nuclease. How specifically the motor of hDNA2 overcomes the inhibition by the hDNA2 nuclease thus remains to be established. Our results show that the helicase of hDNA2 may play a non-essential but stimulatory role in conjunction with BLM or WRN. Previously, the helicase of hDNA2 and its yeast homologue was found dispensable for resection *in vivo* (*Thangavel et al., 2015*; *Zhu et al., 2008*), which contrasts with the results obtained in this study. However, the previous experiments were carried out under conditions where the complementing hDNA2 variants, either wild type or helicase-deficient, were expressed ectopically from a plasmid. This might have masked the stimulatory role of the hDNA2 helicase. Overexpression of wild type yDna2 leads to cell cycle arrest (*Parenteau and Wellinger, 1999*), showing that the levels of DNA2 must be balanced. Further experiments presented here provided evidence that both WRN and BLM promote dsDNA unwinding by hDNA2 and *vice versa*, showing that the helicases of

hDNA2 and WRN/BLM functionally integrate (*Figure 7*). Taken together, our results suggest that the helicase of hDNA2 might play a supportive role in DNA end resection of DNA double-strand breaks, reversed replication forks and/or other structures arising in S phase. The failure and/or delay in the repair of these structures then result in the pronounced G2 phase cell cycle arrest and checkpoint signaling that had been observed in the absence of the hDNA2 helicase (*Duxin et al., 2012*).

Finally, hDNA2 was found to be overexpressed in various human cancers, and the hDNA2 expression level negatively correlated with disease outcome (*Peng et al., 2012*; *Strauss et al., 2014*). This suggested that hDNA2 might especially promote viability of rapidly-dividing cancer cells with high levels of replication stress. This identified hDNA2 as a potential target for anti-cancer therapy. We show that inhibition of the nuclease activity unleashes the hDNA2 helicase, which is likely to contribute to the cytotoxic effects of the hDNA2 nuclease inhibitors. Duxin *et al*. observed that human cells expressing the nuclease-deficient variant were rapidly selected against during the course of the experiment, unlike in case of the double mutant lacking both nuclease and helicase activities that was maintained at constant levels (*Duxin et al., 2012*). Therefore, subsequent inactivation of the helicase might lead to resistance to the hDNA2 nuclease inhibitors. The assays developed in this study will be invaluable to assess the specificity and the mechanism of action of the various hDNA2 inhibitors that are currently being developed.

## Materials and methods

### Preparation of recombinant proteins

The *hDNA2* sequence was codon optimized for the expression in *Sf9* insect cells (*Supplementary file 1A*) and was purchased from GenScript (Piscataway, NJ). The *hDNA2* gene was amplified by PCR using primers 5'-TAGGAAGGATCCATGCATCACCATCACCATCACGGTGGTTC TGGTATGGAGCAATTGAACGAACTCGAAC-3' and 5'-GGTCACAAGCTTTTACTTATCGTCGTCA TCCTTGTAATCTTCACGCTGGAAGTCGCCG-3' to introduce BamHI and HindIII restriction sites as well as 6xHis and FLAG tags (*Figure 1A*). The PCR products were digested with BamHI and HindIII restriction endonucleases (New England Biolabs, Ipswich, MA) and ligated into a pFastBac1 vector (Invitrogen, Carlsbad, CA) generating pFB-His-hDNA2-FLAG. The D277A point mutation inactivating the hDNA2 nuclease was introduced with oligonucleotide pair 5'-GGCCTGAAGGGAAAGATCGCTG TCACAGTTGGAGTGAAG-3' and 5'-CTTCACTCCAACTGTGACAGCGATCTTTCCCTTCAGGCC-3' whereas the K654R point mutation abolishing the hDNA2 helicase was introduced with oligonucleotide pair 5'-GGCATGCCGGGAACTGGCAGGACAACCACTATCTGCACA-3' and 5'-TGTGCAGATAG TGGTTGTCCTGCCAGTTCCCGGCATGCC-3' using the QuikChange XL Site-directed mutagenesis kit (Agilent, Santa Clara, CA) according to manufacturer's recommendations. The construct for the expression of the helicase and nuclease-deficient D277A K654R hDNA2 double mutant was prepared sequentially using the primers described above. All hDNA2 variants were expressed in *Sf9* insect cells in SFX Insect serum-free medium (Hyclone, GE Healtcare, UK) using the Bac-to-Bac expression system (Thermo Fisher Scientific, Waltham, MA), according to manufacturer's recommendations. Frozen *Sf9* pellets from 3 liters culture for each variant were re-suspended in lysis buffer (50 mM Tris-HCl pH 7.5, 2 mM β-mercaptoethanol, 1 mM phenylmethanesulfonylfluoride [PMSF], 1 mM ethylenediaminetetraacetic acid [EDTA], 10 mM imidazole, protease inhibitor cocktail [P8340, Sigma-Aldrich, St. Louis, MO] diluted 1:250, 30 µg/ml leupeptin [Merck Millipore, Billerica, MA]) and incubated at 4°C for 20 min. Glycerol was added to a final concentration of 15%, NaCl was added to a final concentration of 305 mM and the solution was incubated at 4°C for 30 min. The mixture was centrifuged at 39'000 g at 4°C for 30 min. The soluble extract was incubated with Ni-NTA agarose resin (Qiagen, Germany) at 4°C for 1 hr. Ni-NTA resin was washed with Ni-NTA wash buffer 1 M (50 mM Tris-HCl pH 7.5, 2 mM β-mercaptoethanol, 1 mM PMSF, 1 mM EDTA, 10 mM imidazole, 1:1000 protease inhibitor cocktail, 30 µg/ml leupeptin, 10% glycerol, 1 M NaCl) and subsequently washed with Ni-NTA wash buffer 150 mM (the same buffer as above, but only with 150 mM NaCl). Proteins were eluted using Ni-NTA wash buffer 150 mM supplemented with 300 mM imidazole, and subsequently diluted with 4 volumes of FLAG wash buffer 150 mM (50 mM Tris-HCl pH 7.5, 0.5 mM β-mercaptoethanol, 1 mM PMSF, 10% glycerol, 150 mM NaCl) to lower the imidazole and β-mercaptoethanol concentrations. The mixture was incubated with anti-FLAG M2 Affinity Gel (A2220, Sigma-Aldrich) at 4°C for 1 hr. Proteins were eluted using FLAG wash buffer 150 mM supplemented with

300 µg/ml FLAG peptide (F4799, Sigma-Aldrich), aliquoted, snap-frozen in liquid nitrogen and stored at −80°C. Yeast Dna2, Sgs1, human and yeast RPA were purified as described previously (*Cejka and Kowalczykowski, 2010*; *Henricksen et al., 1994*; *Kantake et al., 2003*; *Levikova et al., 2013*). Human RecQ1, RecQ5 and yeast Srs2 and their variants were kind gifts from A. Vindigni (Saint Louis University, USA), P. Janscak (University of Zurich, Switzerland) and L. Krejci (Masaryk University, Czech Republic).

The *BLM* gene was amplified by PCR from pZL4 plasmid (*Kanagaraj et al., 2006*) with primers 5'-TAGGAAGCTAGCGGATCCATGGCTGCTGTTCCTCAAAA-3' and 5'-TAGGAACTCGAGCCCGGG TGAGAATGCATATGAAGGCTT-3' to introduce XhoI and NheI restriction sites. The *WRN* gene was amplified by PCR from plasmid pBlueBacHis-WRN (*Gray et al., 1997*) with primers 5'-TAGGAAGC TAGCGGATCCATGAGTGAAAAAAAATTGGAAACAA-3' and 5'-TAGGAACTCGAGCCCGGGAC TAAAAAGACCTCCCCTTTT-3' to introduce XhoI and NheI restriction sites. The PCR products were cloned into pFB-MBP-Sgs1-his (*Cejka et al., 2010*) generating pFB-MBP-BLM-his and pFB-MBP-WRN-his, respectively. Mutations for the helicase-deficient variants were introduced as described above using oligonucleotide pairs 5'-ACTGGAGGTGGTGCGAGTTTGTGTTACCAGCTC-3' and 5'-GAGCTGGTAACACAAACTCGCACCACCTCCAGT-3' for BLM K695A and 5'-GCAACTGGATA TGGAATGAGTTTGTGCTTCCAGTATCC-3' and 5'-GGATACTGGAAGCACAAACTCATTCCATA TCCAGTTGC-3' for WRN K577M. All BLM and WRN variants were expressed in *Sf*9 cells. Frozen *Sf*9 pellets from 1.2–2 l culture for each variant were resuspended in lysis buffer and soluble extract was prepared as for hDNA2 (see above). The soluble extract was incubated with amylose resin (New England Biolabs) at 4°C for 1 hr. The resin was washed with amylose wash buffer 1 M (50 mM Tris-HCl pH 7.5, 5 mM β-mercaptoethanol, 1 mM PMSF, 10% glycerol, 1 M NaCl). Proteins were eluted using amylose elution buffer (50 mM Tris-HCl pH 7.5, 5 mM β-mercaptoethanol, 1 mM PMSF, 10% glycerol, 300 mM NaCl, 10 mM maltose). The MBP-tagged variants were incubated with PreScission protease (~25 µg PreScission protease per 100 µg of tagged protein) at 4°C for 1.5 hr to cleave the MBP tag. Subsequently, imidazole was added to a final concentration of 10 mM and the solution was incubated with pre-equilibrated Ni-NTA agarose resin (Qiagen) at 4°C for 1 hr, in agitation. The resin was washed with NTA Buffer A1 (50 mM Tris-HCl pH 7.5, 5 mM β-mercaptoethanol, 1 mM PMSF, 10% glycerol, 1 M NaCl, 58 mM imidazole) and subsequently with NTA Buffer A2 (50 mM Tris-HCl pH 7.5, 5 mM β-mercaptoethanol, 1 mM PMSF, 10% glycerol, 150 mM NaCl, 58 mM imidazole). The BLM or WRN variants were eluted with NTA Buffer B (50 mM Tris HCl pH 7.5, 5 mM β-mercaptoethanol, 1 mM PMSF, 10% glycerol, 100 mM NaCl, 300 mM imidazole). Fractions containing high protein concentration were pooled and dialyzed against 1 l of dialysis buffer (50 mM Tris-HCl pH 7.5, 5 mM β-mercaptoethanol, 0.5 mM PMSF, 10% glycerol, 100 mM NaCl) for 1 hr at 4°C. Proteins were aliquoted, snap-frozen in liquid nitrogen and stored at −80°C. The typical yield was ~50–280 µg from 1.2–2 liters culture for each variant.

The plasmid pSF1-hsmtSSB coding for human mitochondrial single-stranded DNA binding protein (mtSSB) was received from Ute Curth (Hannover Medical School) (*Curth et al., 1994*). The mtSSB gene was amplified by PCR using primers 5'-GTGACCGAATCATGGACTCCGAAACAACTACCAG TTTGG-3' and 5'-GTGACCGGATCCCTACTCCTTCTCTTTCGTCTGGTCACTC-3' and cloned into the pMALT-P expression vector (Taeho Kim, Kowalczykowski laboratory, unpublished) using EcoRI and BamHI restriction sites. This placed mtSSB behind an MBP tag and a PreScission Protease site creating pMALT-P-mtSSB. The MBP-mtSSB fusion was expressed in *E. coli* BL21 cells upon induction with Isopropyl β-D-1-thiogalactopyranoside (IPTG, 400 µM) for 3 hr at 37°C. Frozen *E. coli* pellets from 2 l *E. coli* culture were re-suspended in buffer B1 (50 mM Tris-HCl pH 7.5, 1 mM EDTA, protease inhibitor cocktail [1:400], 30 µg/ml leupeptin, 1 mM PMSF, 1 mM dithiothreitol [DTT], 10% glycerol, 100 mM NaCl) and lysed by sonication. Whole cell extract was centrifuged at 39'000g at 4°C for 30 min. The supernatant was collected and incubated with pre-equilibrated 3 ml amylose resin (New England Biolabs) for 1 hr at 4°C. The resin was washed in buffer B2 (50 mM Tris-HCl pH 7.5, 1 mM DTT, 10% glycerol, 100 mM NaCl). MBP-mtSSB was eluted using buffer B3 (50 mM Tris-HCl pH 7.5, 1 mM DTT, 10% glycerol, 100 mM NaCl, 10 mM maltose). The MBP tag was cleaved with PreScission protease (~15 µg per 100 µg MBP-mtSSB) overnight at 4°C. The solution was then diluted with 1 volume of water. The sample was loaded onto a HiTrap Blue column (GE Healthcare). The column was washed with buffer B2 sequentially supplemented with 50 mM KCl, 800 mM KCl, 0.5 M sodium thiocyanate and 1.5 M sodium thiocyanate. The mtSSB protein was eluted using buffer B4 (20 mM Tris-HCl pH 7.5, 1 mM EDTA, 1 mM DTT, 10% glycerol, 2 M NaCl, 5 M urea). The eluate was dialyzed

twice against 1 l buffer B5 (50 mM Tris-HCl pH 7.5, 1 mM DTT, 10% glycerol, 50 mM NaCl) for 1.5 hr each time. The final mtSSB preparation was aliquoted, snap-frozen in liquid nitrogen and stored at −80°C.

The three pDONR plasmids (Thermo Fisher Scientific) coding for SOSSA, SOSSB1 and SOSSC, respectively, were gifts from Jun Huang (Life Sciences Institute, Hangzhou, Zhejiang University) (Huang et al., 2009). The *SOSSA* gene was cloned into pDEST20 vector using Gateway recombination cloning technology (Thermo Fisher Scientific) creating pDEST20-GST-SOSSA. To add a N-terminal 6xhis tag to SOSSB1, the *SOSSB1* gene was amplified by PCR using primers 5'-GTGACCGGA TCCATGCATCACCATCACCATCACATGACGACGGAGACCTTTGTGAAGGATATC-3' and 5'-G TGACCCCCGGGCTATCTCTTGCTGCTCCTCCGGGTTT-3'. The PCR product was cloned into a pFastBac1 vector using BamHI and XmaI restriction sites, creating pFB-hisSOSSB1. The *SOSSC* gene was amplified by PCR using primers 5'-GTGACCGGATCCATGGCAGCAAACTCTTCAGGACAAGG TTTTC-3' and 5'-GTGACCC'CCGGGTCATTCTGGGTCAAGGCGAGGTAAAACAG-3'. The gene was cloned into pFastBac1 vector using BamHI and XmaI restriction sites, creating pFB-SOSSC. The heterotrimer was expressed as a complex in *Sf*9 cells for in 2 l of culture. The pellet was re-suspended in buffer B1 (50 mM Tris-HCl pH 8, 1 mM EDTA, protease inhibitor cocktail [1:400], 30 μg/ml leupeptin, 1 mM PMSF, 1 mM DTT) and incubated for 20 min at 4°C in agitation. Glycerol was added to a final concentration of 15% and NaCl was added to a final concentration of 305 mM. The solution was incubated for another 30 min at 4°C in agitation. The solution was centrifuged for 30 min at 39'000 g at 4°C. The soluble extract was then incubated with 2 ml pre-equilibrated Glutathione HiCap matrix (Qiagen) for 1 hr at 4°C in agitation. The resin was washed 3x batch-wise and subsequently on column with wash buffer (50 mM Tris-HCl pH 8, protease inhibitor [1:1000], 1 mM PMSF, 2 mM 2-mercaptoethanol). The proteins were eluted using the wash buffer supplemented with 10 mM glutathione. Imidazole was added to the eluate to a final concentration of 10 mM and incubated with pre-equilibrated Ni-NTA agarose resin (Qiagen) for 1 hr at 4°C in agitation. The resin was washed twice with buffer A2 (50 mM Tris HCl pH 7.5, 2 mM 2-mercaptoethanol, 150 mM NaCl, 10% glycerol, 1 mM PMSF, 58 mM imidazole) and the proteins were eluted with buffer B (50 mM Tris-HCl pH 7.5, 2 mM 2-mercaptoethanol, 100 mM NaCl, 10% glycerol, 1 mM PMSF, 300 mM imidazole). The eluate was dialyzed against 1 l of dialysis buffer (50 mM Tris-HCl pH 7.5, 2 mM 2-mercaptoethanol, 100 mM NaCl, 10% glycerol, 0.5 mM PMSF) for 1.5 hr at 4°C.

## DNA substrates

Oligonucleotides were labeled either at the 5' terminus with $[\gamma\text{-}^{32}P]$ATP and T4 polynucleotide kinase (New England Biolabs), or at the 3' terminus with $[\alpha\text{-}^{32}P]$ cordycepin-5-triphosphate and terminal transferase (New England Biolabs) according to standard protocols. The sequences of all oligonucleotides are listed in *Supplementary file 1B*. The substrates were prepared by annealing the $^{32}P$-labeled oligonucleotide with a two-fold excess of the unlabeled oligonucleotide in a PNK buffer (New England Biolabs). The substrates and component oligonucleotides are listed in *Supplementary file 1C*.

λDNA/HindIII fragments (Bacteriophage λ DNA-HindIII Digest, New England Biolabs) were labeled at the 3' ends with $[\alpha\text{-}^{32}P]$dATP and Klenow fragment of DNA polymerase I (New England Biolabs) in NEBuffer 2. The pUC19 plasmid was digested by HindIII-HF restriction enzyme and purified by phenol-chloroform extraction and ethanol precipitation. The resulting linear dsDNA was labeled at the 3' ends with $[\alpha\text{-}^{32}P]$dATP and Klenow fragment of DNA polymerase I (New England Biolabs) in NEBuffer 2. Unincorporated radioactive ATP was in all cases removed using Micro Spin G25 columns (GE Healthcare). The positions of the radioactive labels are indicated in the substrate schematics with a star symbol.

## Electrophoretic mobility shift assays

The reactions (15 μl volume) were performed in a binding buffer (25 mM Tris-acetate pH 7.5, 2 mM magnesium acetate, 1 mM DTT, 0.1 mg/ml BSA) with the respective DNA substrate (1 nM). The reactions were incubated at 37°C for 30 min. Loading dye (50% glycerol, bromophenol blue) was added and the products were separated by polyacrylamide gel electrophoresis (6%, ratio acrylamide-bisacrylamide 19:1, BioRad) in Tris-Acetate-EDTA (TAE) buffer. The electrophoresis was carried out in a gel tank surrounded by ice. The gels were dried on DE81 chromatography paper (Whatman, UK).

The dried gels were then exposed to Storage Phosphor screens (GE Healthcare) and scanned by Typhoon 9400 (GE Healthcare). The data was quantified using Image Quant TL software (GE Healthcare).

## Nuclease assays

Nuclease assays (15 µl volume) were performed in a reaction buffer (25 mM Tris-acetate pH 7.5, 2 mM magnesium acetate, 1 mM ATP, 1 mM DTT, 0.1 mg/ml BSA, 1 mM phosphoenolpyruvate (PEP), 0.02 units/µl pyruvate kinase [Sigma]) containing DNA substrates (1 nM) and recombinant proteins as indicated. Reactions were incubated at 37°C for 30 min. For analysis on native gels the reactions were stopped by adding 5 µl 2% stop solution (150 mM EDTA, 2% sodium dodecyl sulfate [SDS], 30% glycerol, bromophenol blue) and 1 µl Proteinase K (14-22 mg/ml, Roche, Switzerland) and incubated at 37°C for 10 min. The samples were then analyzed by native polyacrylamide gel electrophoresis (10%, ratio acrylamide-bisacrylamide 19:1, Biorad, Hercules, CA). For analysis on denaturing gels the reactions were stopped by adding an equal amount of formamide dye (95% [v/v] formamide, 20 mM EDTA, bromophenol blue), samples were heated at 95°C for 4 min and separated on 20% denaturing polyacrylamide gels (ratio acrylamide:bisacrylamide 19:1, Biorad). After fixing in a solution containing 40% methanol, 10% acetic acid and 5% glycerol for 30 min the gels were dried and analyzed as described above.

## Helicase and ATPase assays

Helicase assays (15 µl volume) were performed in a reaction buffer (25 mM Tris-acetate pH 7.5, 2 mM magnesium acetate, 1 mM ATP, 1 mM DTT, 0.1 mg/ml BSA, 1 mM PEP, 0.02 units/µl pyruvate kinase) with the respective DNA substrate (1 nM for oligonucleotide- and pUC19/HindIII-based and 0.15 nM for λDNA/HindIII-based substrates). Recombinant proteins were added as indicated. Reactions were incubated at 37°C for 30 min and stopped as described above in the nuclease assay section. To avoid re-annealing of the oligonucleotide-based substrates, the stop solution was supplemented with a 20-fold excess of the oligonucleotide with the same sequence as the $^{32}$P-labeled one. The products were analyzed either by polyacrylamide gel electrophoresis (10%) for oligonucleotide-based DNA substrates or 1% agarose gels for plasmid and λDNA-based DNA substrates. The gels were dried on DE81 chromatography paper (Whatman) and analyzed as described above. The ATPase assays were performed as described previously (*Kowalczykowski and Krupp, 1987*). The reaction buffer contained 25 mM Tris-acetate pH 7.5, 1 mM magnesium acetate, 1 mM DTT, 0.1 mg/ml BSA, 1 mM ATP and 1 mM PEP, 0.025 units/µl pyruvate kinase, 0.025 units/µl L-lactic dehydrogenase (Sigma).

## Magnetic tweezers assay

The DNA construct was prepared as described before (*Levikova et al., 2013*). The central part is a 6.1 kbp dsDNA having a ssDNA flap of 40 nt in length placed 1 kbp from its proximal DNA end. A digoxigenin- and a biotin-modified handle of 600 bp length were attached to the 6.1 kbp fragment at its flap-proximal and distal ends, respectively. The magnetic tweezers experiments were carried out as described previously (*Klaue and Seidel, 2009*; *Levikova et al., 2013*). In brief, the DNA substrate was bound to 2.8 µm streptavidin-coated magnetic beads (M280, Invitrogen) and were flushed into the fluidic cell, whose bottom glass slide was covered with digoxigenin. After a brief incubation to allow the attachment of the digoxigenin-modified DNA end, a pair of magnets above the flow cell was approached to remove unbound beads and to stretch the bead tethered DNA molecules. The DNA length was obtained by videomicroscopy of the beads and GPU-accelerated real-time particle tracking (*Huhle et al., 2015*). The stretching force was adjusted by changing the distance of the magnet to the fluidic cell. Forces were calibrated for each bead using fluctuation analysis (*Daldrop et al., 2015*). The unwinding assays were performed in a reaction buffer (25 mM Tris-acetate pH 7.5, 5 mM magnesium acetate, 1 mM ATP, 1 mM DTT, 0.1 mg/ml BSA) supplemented with 25 nM hRPA and 25 nM hDNA2 D277A at 37°C. For temperature control of the setup an objective heater (Okolab, Pozzuoli, Italy) was employed. With respect to the bulk assays, the magnesium acetate concentration was increased to 5 mM to prevent DNA melting by hRPA (*Kemmerich et al., 2016*). The DNA extension resulting from unwinding was converted from µm into bp applying a

conversion factor that was calculated from the force extension curves for the DNA molecule at the certain force (*Kemmerich et al., 2016*).

## Acknowledgements

We would like to thank Pavel Janscak (University of Zurich), Ute Curth (Hannover Medical School), Jun Huang (Zhejiang University), Lumir Krejci (Masaryk University) and Alessandro Vindigni (Saint Louis University) for recombinant proteins and expression constructs. We thank Andrey Krivoy for helping establish the temperature control within the magnetic tweezers. We thank Elda Cannavo, Maryna Levikova, Roopesh Anand and Lucie Mlejnkova for critical reading of the manuscript. This work was supported by the Swiss National Science foundation grant PP00P3 159323 to PC, Swiss Cancer League grant KFS-3089-02-2013 to PC and ERC starting grant (GA 261224) to RS.

## Additional information

### Funding

| Funder | Grant reference number | Author |
| --- | --- | --- |
| Schweizerischer Nationalfonds zur Förderung der Wissenschaftlichen Forschung | PP00P3 159323 | Petr Cejka |
| Krebsliga Schweiz | KFS-3089-02-2013 | Petr Cejka |
| European Research Council | GA 261224 | Ralf Seidel |

The funders had no role in study design, data collection and interpretation, or the decision to submit the work for publication.

### Author contributions

CP, Conception and design, Acquisition of data, Analysis and interpretation of data, Drafting or revising the article; KK, Acquisition of data, Analysis and interpretation of data, Drafting or revising the article; RS, PC, Conception and design, Analysis and interpretation of data, Drafting or revising the article

### Author ORCIDs

Petr Cejka, http://orcid.org/0000-0002-9087-032X

## Additional files

### Supplementary files

• Supplementary file 1. DNA sequences used in this study. (A) Codon-optimized nucleotide sequence of hDNA2 gene for the expression in *Sf9* cells. (B) Sequences of oligonucleotides used in this study. (C) Oligonucleotide-based DNA substrates used in this study.

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
