## [Decision Letter]

Thank you for submitting your article "Human DNA2 possesses a cryptic DNA unwinding activity that functionally integrates with BLM or WRN helicases" for consideration by *eLife*. Your article has been favorably evaluated by Jessica Tyler as the Senior Editor and three reviewers, including Maria Spies and a member of our Board of Reviewing Editors.

The reviewers have discussed the reviews with one another and the Reviewing Editor has drafted this decision to help you prepare a revised submission.

Summary:

The authors describe helicase activity of the human DNA2 protein, thereby reconciling a large body of literature with conflicting evidence on the existence of such activity and the interplay with DNA2's nuclease activity. They use a combination of biochemical and single-molecule approaches to measure the basic kinetic properties of the helicase activity and show synergistic effects with the BLM and WRN factors in end resection.

The reviewers express excitement about the findings that human DNA2 possesses a cryptic DNA helicase activity, normally masked by its nuclease activity, and that hDNA2 can work together with Werner and Bloom helicases to promote degradation of double-strand DNA ends. However, they also raise significant concerns about the evidence supporting the latter claim. The revisions required to address these concerns will necessitate the introduction of further experimental evidence, as described below.

Essential revisions:

The reviewers' main concern is that the authors have not demonstrated convincingly a specific stimulatory effect of WRN or BLM (assuming that this is what they mean when they describe how human DNA2 'acts synergistically' or 'functionally integrate' with WRN and BLM). They have simply shown that the inhibition of DNA2's helicase activity by its nuclease activity can be obviated by adding a helicase in trans. The authors will need to provide a critical negative control, by showing that the stimulation of DNA degradation by DNA2 in the presence of WRN and BLM is greater than that provided by a functionally-unrelated DNA helicase.

As they support a central conclusion of the paper, the experiments of Figure 6 and Figure 7 should be extended. They should include a titration of WRN, BLM and negative control in the DNA2 helicase assay, using D277A DNA2 (the reciprocal titration of current 6A, B), and a time course of DNA2 unwinding in the presence of fixed amounts of WRN, BLM and negative control. As they presumably act together during processing of the DNA substrate, the order of addition of the various helicases should also be tested. Finally, the authors may want to consider measuring the effect of WRN and BLM on the rates of DNA2 unwinding using the single-particle setup, providing what probably is the best evidence of stimulation.

---

## [Author Response]

*The reviewers' main concern is that the authors have not demonstrated convincingly a specific stimulatory effect of WRN or BLM (assuming that this is what they mean when they describe how human DNA2 'acts synergistically' or 'functionally integrate' with WRN and BLM). They have simply shown that the inhibition of DNA2's helicase activity by its nuclease activity can be obviated by adding a helicase in trans. The authors will need to provide a critical negative control, by showing that the stimulation of DNA degradation by DNA2 in the presence of WRN and BLM is greater than that provided by a functionally-unrelated DNA helicase.*

*As they support a central conclusion of the paper, the experiments of Figure 6 and Figure 7 should be extended. They should include a titration of WRN, BLM and negative control in the DNA2 helicase assay, using D277A DNA2 (the reciprocal titration of current 6A, B), and a time course of DNA2 unwinding in the presence of fixed amounts of WRN, BLM and negative control. As they presumably act together during processing of the DNA substrate, the order of addition of the various helicases should also be tested. Finally, the authors may want to consider measuring the effect of WRN and BLM on the rates of DNA2 unwinding using the single-particle setup, providing what probably is the best evidence of stimulation.*

We agree with these comments. First, we clarified in the text that by "synergy" we mean a species-specific stimulatory effect. We extended the data presented in Figure 6 and Figure 7 and associated Supplements by adding the following new experiments:

Figure 6 now include the vice versa titration related to Figure 6 using fixed amounts of hDNA2 and titrating increasing amounts of WRN or BLM helicase into the reaction, as requested. Again we could see the stimulatory effect of the enzyme pair, i.e. adding hDNA2 to the helicase resulted in more DNA degradation than the amount of DNA unwound by the helicase alone.

Figure 6 now shows the critical specificity control, where adding hDNA2 to the yeast RecQ family helicase Sgs1 (related to BLM and WRN) resulted only in the degradation DNA unwound by Sgs1, and no substrate degradation enhancement was observed. It even appears that the overall DNA unwinding by Sgs1 was inhibited by hDNA2, resulting in less DNA substrate utilized. This experiment demonstrates the specific nature of the stimulatory effect of hDNA2 helicase-nuclease together with either WRN or BLM helicases in DNA degradation. Furthermore, Figure 6—figure supplement 1 shows that using RecQ1 and RecQ5 helicases in a similar setup did not show the synergistic effect, indicating that the stimulation is specific for WRN and BLM but not for other helicases from the human RecQ family.

Figure 6 show the requested time course experiments with fixed amounts of hDNA2 and WRN or BLM helicases as well as the negative control Sgs1. We detected a rapid degradation of the dsDNA by the hDNA2-WRN complex followed by the hDNA2-BLM complex, showing a pronounced stimulatory effect in both cases. The synergy was in contrast not observed in experiments using hDNA2 and yeast Sgs1 confirming the species-specific nature of the stimulatory effect. The results are then quantified in panel 6J.

To extend the data presented in Figure 7, we tested whether helicase-deficient mutants of various yeast and human DNA helicases could stimulate the helicase of hDNA2 D277A similarly as WRN or BLM helicase-deficient mutants as shown in Figure 7. However, we could not observe such a stimulation (Figure 7—figure supplement 1), showing that the effect is specific for WRN and BLM helicases.

We would like to kindly ask the reviewers to consider the synergy experiments in the single-molecule setup as above the scope of the current manuscript. We however believe that the additional bulk experiments presented in the revised manuscript demonstrate the specificity issue convincingly.

We observed the most consistent results when both enzymes were pre-mixed before adding to the substrate. We observed the stimulatory effect as well when the enzymes were added to the substrate at various orders of addition, however the experiments were less reproducible, probably because insufficient time for a complex formation and/or too low protein concentrations for efficient complex formation.